# Unlocking the matrix form of the Quaternion Fourier Transform and Quaternion Convolution: Properties, connections, and application to Lipschitz constant bounding

**Giorgos Sfikas**                                                                 *gsfikas@uniwa.gr*
*Department of Surveying and Geoinformatics, School of Engineering*
*University of West Attica*
*Athens, Greece*

**George Retsinas**                                                            *gretsinas@central.ntua.gr*
*School of Electrical and Computer Engineering*
*National Technical University of Athens*
*Athens, Greece*

**Reviewed on OpenReview:** *https://openreview.net/forum?id=rhcpXTxb8j*

## Abstract

Linear transformations are ubiquitous in machine learning, and matrices are the standard way to represent them. In this paper, we study matrix forms of quaternionic versions of the Fourier Transform and Convolution operations. Quaternions offer a powerful representation unit, however they are related to difficulties in their use that stem foremost from non-commutativity of quaternion multiplication, and due to that $\mu^2 = -1$ possesses infinite solutions in the quaternion domain. Handling of quaternionic matrices is consequently complicated in several aspects (definition of eigenstructure, determinant, etc.). Our research findings clarify the relation of the Quaternion Fourier Transform matrix to the standard (complex) Discrete Fourier Transform matrix, and the extend on which well-known complex-domain theorems extend to quaternions. We focus especially on the relation of Quaternion Fourier Transform matrices to Quaternion Circulant matrices (representing quaternionic convolution), and the eigenstructure of the latter. A proof-of-concept application that makes direct use of our theoretical results is presented, where we present a method to bound the Lipschitz constant of a Quaternionic Convolutional Neural Network. Code is publicly available at: https://github.com/sfikas/quaternion-fourier-convolution-matrix.

## 1 Introduction

Quaternions are four-dimensional objects that can be understood as generalizing the concept of complex numbers. Some of the most prominent applications of quaternions are in the fields of photogrammetry (Förstner & Wrobel, 2016), computer graphics (Vince, 2011), robotics (Daniilidis, 1999; Fresk & Nikolakopoulos, 2013) and quantum mechanics (Adler, 1995; Susskind & Friedman, 2014). Perhaps the most well known use case of quaternions involves representing rotations in three-dimensional space (Kuipers, 1999; Stillwell, 2008; Eater & Sanderson, 2018). This representation is convenient, as for example a composition of rotations corresponds to a multiplication of quaternions, and both actions are non-commutative. Compared to other spatial rotation representations such as Euler angles, quaternions are advantageous in a number of respects (Förstner & Wrobel, 2016).

A relatively more recent and perhaps overlooked use of quaternionic analysis traces its roots in digital signal & image processing (Sangwine, 1996; Sangwine & Le Bihan, 2006; Subakan & Vemuri, 2011; Grigoryan & Agaian, 2014; Miron et al., 2023). The motivation for using quaternionic extensions of standard theoretical

signal processing tools such as filtering and the Fourier Transform (FT), is that their most typical application on color images involves treating them as three separate monochrome images. As a consequence, cross-channel dynamics are ignored in this approach. The solution is to treat each color value as a single, unified object, and one way to do so is by representing multimodal pixel values as quaternions. The caveat of this approach would be that standard tools, filters and methods would have to be redefined and remodeled, a process that is typically not always straightforward (a fact that partly motivates this work). Similar considerations hold for other types of signals, such as acoustic signals or polarimetric imaging (Miron et al., 2023). Recent works move away from the scope set originally by such physically-based motivations and extend to other applications, including using quaternion-valued elements as components of Deep Neural Networks. Part of the network may be expressed in terms of quaternions (Rosa et al., 2018; Hsu et al., 2019; Qin et al., 2023) or even all components of the network –inputs, intermediate layers, outputs– as done in Quaternionic Neural Networks. The latter lead in practice to models that can incorporate quaternionic versions of standard layers such as convolution or self-attention, have much less memory requirements than their real-valued counterparts, and perform similarly in terms of generalization (Zhu et al., 2018; Parcollet et al., 2020; Zhang et al., 2021). Parameterized hypercomplex layers build on this paradigm (Zhang et al., 2021), and neural networks can be effectively extended to work with elements of an arbitrary number of dimensions.

In this paper, our focus is on the quaternionic version of the Discrete Fourier transform (DFT) and its relation to Quaternionic Convolution. The Quaternion Fourier Transform (QFT) was originally proposed by Sangwine (1996), and in effect it provides the means to analyze and manipulate the frequency content of multichannel signals. We argue that, while the QFT has been studied by a number of works and has found uses in practical applications (Ell & Sangwine, 2007; Subakan & Vemuri, 2011; Li et al., 2015; Hitzer, 2016), its potential has yet to be exploited in its fullest.

From another standpoint, we can frame our work as serving to elucidate the properties of Convolution in $\mathbb{H}$. We contribute to the better understanding of the convolutional operator, and provide the means to analyze and manipulate its spectral properties. Convolution is a ubiquitous operation in data science and machine learning, and its instance in the quaternion domain forms a challenging, yet interesting and promising tool that is hitherto underexplored. The extensive use and long history of convolution in the context of neural networks is very well-known (Prince, 2023); while it can be argued that the field is rapidly evolving with non-convolutional backbones such as Transformers or State-Space models (Gu & Dao, 2023), the intrinsic connection of convolution with notions such as invariance and equivariance (Kondor & Trivedi, 2018) or the already known connection to frequency analysis in $\mathbb{R}$ (Jain, 1989) (and extended with the current work in $\mathbb{H}$), are strong pointers in favor of the continued relevance of convolution. The success of hybrids of non-convolution with convolution or convolutional components seems to corroborate this point (Wick et al., 2022; Zhang et al., 2019; Xu et al., 2023; Gu & Dao, 2023; Yu & Wang, 2025). Our work serves to further widen the usefulness and domain of application of convolution.

A matrix representation form for the Quaternion Fourier Transform and Convolution and their properties are studied in this paper [1]. We show that a number of properties of the DFT matrix form also do extend to the quaternionic domain, in most cases appearing as a more complex version of the original (DFT) property. We focus on its relation to Quaternionic Convolution and its circulant form, extend the well-known results for the non-quaternionic case (Jain, 1989), and show that the eigenstructure of Quaternionic Circulant matrices is closely connected to the Quaternion Fourier Matrix form.

The major contributions of this paper are the following:

1. We show that convolutions and Fourier transforms in $\mathbb{H}$ are represented as matrices with properties that are analogous to their real and complex-domain counterparts; the standard (complex) Fourier matrix can be seen a special case out of an infinite set of Quaternionic Fourier matrices (cf. Propositions 3.1-3.3), which are related through a rotation-like operation (cf. Prop. 3.4).

---

[1]Concerning prior work, to our knowledge only (Ell et al., 2014, sec.3.1.1.1) refer to a matrix form of the QFT, without proving or discussing any of its properties, its relation to convolution or their implications.

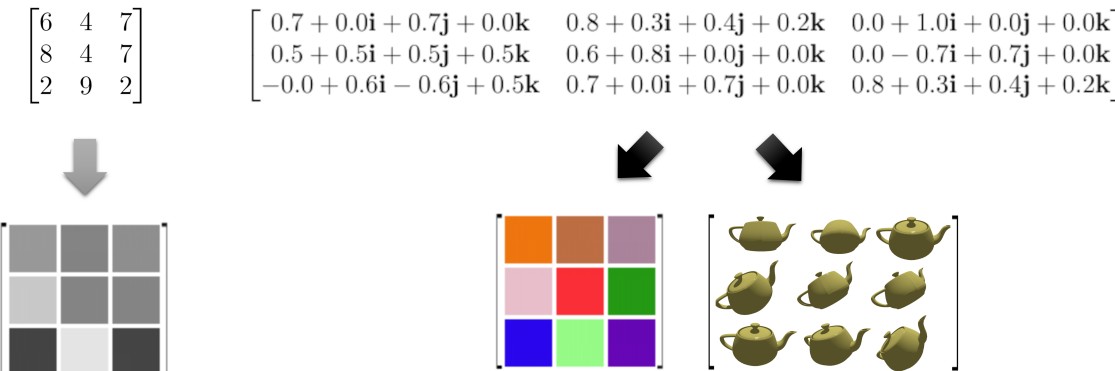

Figure 1:    Matrices of real-valued elements (top left) form a standard representational unit for unidimensional data, such as grayscale pixels (bottom left), neural network weights or other unidimensional signals. Matrices of quaternion-valued elements (top right) represent an arrangement of multidimensional data such as color images (Ell & Sangwine, 2007) or 3D spatial rotation (Kerbl et al., 2023) (bottom right).

2. We show that circulant, doubly block-circulant matrices and Fourier matrices are "still" related in terms of their properties also in the quaternionic domain, *albeit in a more nuanced manner.* In our opinion, it is notable that both the *left* and *right* QFT provide us with different functions of the convolution kernel *left* eigenvalues (cf. Proposition 3.5). This is a result with further theoretical and practical implications, which we discuss (cf. Corollary 3.6.2, as an example).

3. The literature on the theory and applications of quaternion matrices focuses on *right* eigenvalues, as the *left* spectrum is more difficult or impossible to compute in general. Due to our results (Proposition 3.5), our focus must turn to left eigenvalues [2], which we show how to compute using the QFT; we show that a set of $N$ left eigenvalues (out of an infinite number of left eigenvalues of a $N \times N$-sized matrix) can be used to reconstruct a Circulant matrix, under specific conditions (cf. Proposition 3.6).

4. We show that sums, multiples and products of quaternionic circulant matrices can be reconstructed using manipulations in the left spectrum (cf. Proposition 3.7).

5. In terms of application, we show that our results find direct use, replacing and generalizing uses of standard real/complex operators (cf. Section 4). Two-dimensional convolution kernels are shown to be represented as doubly block-circulant constructions (cf. Propositions 3.8,3.9). We present a proof-of-concept application that makes use of our results. Specifically, we propose a way to a) compute the spectral norm of Quaternionic Convolution and b) set a bound on the Lipschitz constant of a Quaternionic Neural Network. These operations are shown to be carried out orders-of-magnitude faster than an implementation that is oblivious to the presented results. This involves a special construction of a matrix that contains $2 \times 2$ blocks of simplex and perplex parts of left eigenvalues (cf. Proposition 4.4, Corollary 4.4.1). This generalizes elegantly from $\mathbb{C}$ to $\mathbb{H}$, as perplex parts vanish in $\mathbb{C}$.

The paper is structured as follows. We proceed to review preliminaries on quaternions and quaternion matrices in Section 2. This includes a short discussion concerning difficulties handling quaternion matrices and therefore operator matrix forms, and outline the net benefits from such a process. In Section 3, we examine the matrix form of the QFT and Quaternion Convolution and present our theoretical contributions regarding their properties. We showcase the usefulness of the discussed form and its properties in Section 4. We conclude the main paper with Section 5. In the Appendix, we have moved supplemental results, a

---

[2]Most hitherto works focus on right eigenvalues; as Aslaksen (2001) notes, "In general, it is difficult to talk about eigenvalues of a quaternionic matrix. Since we work with right vector spaces, we must consider right eigenvalues."

short section on our notation conventions, a section on preliminaries on Quaternion algebra, and additional details on models and proofs.

## 2 Preliminaries on Quaternions and Quaternion Matrices

In this Section, we shall attempt to outline a selection of important definitions, results and key concepts concerning quaternions (Alfsmann et al., 2007; Cheng & Kou, 2016; Ell & Sangwine, 2007; Ell et al., 2014; Le Bihan, 2017), and also provide a concise outlook regarding the pros and cons of their uses in practice.

### 2.1 Elements of Quaternions

The structure $\mathbb{H}$ represents the division algebra that is formed by quaternions. Quaternions share a position of special importance compared to other algebraic structures, as according to Frobenius' theorem (Fraleigh, 2002), every finite-dimensional (associative) division algebra over $\mathbb{R}$ must be isomorphic either to $\mathbb{R}$, or to $\mathbb{C}$, or to $\mathbb{H}$. Quaternions $q \in \mathbb{H}$ share the following basic form:

$$q = a + b\boldsymbol{i} + c\boldsymbol{j} + d\boldsymbol{k}, \tag{1}$$

where $a, b, c, d \in \mathbb{R}$ and $\boldsymbol{i}, \boldsymbol{j}, \boldsymbol{k}$ are independent imaginary units. Real numbers can be regarded as quaternions with $b, c, d = 0$, and complex numbers can be regarded as quaternions with $c, d = 0$. Quaternions with zero real part, i.e. $a = 0$, are called pure quaternions. An alternative way to represent quaternions, is by writing their real and collective imaginary part separately. In particular:

$$q = S(q) + V(q), \tag{2}$$

where $S(q) = a$ and $V(q) = b\boldsymbol{i} + c\boldsymbol{j} + d\boldsymbol{k}$. For all three imaginary units $\boldsymbol{i}, \boldsymbol{j}, \boldsymbol{k}$, it holds $\boldsymbol{i}^2 = \boldsymbol{j}^2 = \boldsymbol{k}^2 = -1$. The length or magnitude of a quaternion is defined as $|q| = \sqrt{q\bar{q}} = \sqrt{\bar{q}q} = \sqrt{a^2 + b^2 + c^2 + d^2}$, where $\bar{q}$ is the conjugate of $q$, defined as $\bar{q} = a - b\boldsymbol{i} - c\boldsymbol{j} - d\boldsymbol{k}$. Quaternions with $|q| = 1$ are named unit quaternions. For any unit pure quaternion $\mu$, the property $\mu^2 = -1$ holds. Exponentials of quaternions $e^x$ for $x \in \mathbb{H}$ can be defined through their Taylor series:

$$e^x = \sum_{n=0}^{\infty} \frac{x^n}{n!}. \tag{3}$$

From eq. 3, Euler's identity extends for quaternions, for $\mu$ unit and pure:

$$e^{\mu\theta} = cos\theta + \mu \sin\theta. \tag{4}$$

Note also the caveat that in general $e^{\mu+\lambda} \neq e^{\mu}e^{\lambda}$. In particular, for $\alpha, \beta \in \mathbb{R}^+$ and two distinct pure unit quaternions $\mu, \nu$, we have $e^{\nu\alpha}e^{\mu\beta} \neq e^{\nu\alpha+\mu\beta}$. Equality holds however, when quaternions $\mu, \nu$ commute. Hence,

$$e^{\mu\alpha}e^{\mu\beta} = e^{\mu\alpha+\mu\beta}, e^{\mu\alpha}e^{\beta} = e^{\mu\alpha+\beta}, \tag{5}$$

since real numbers commute with any quaternion. Furthermore, any quaternion can be written in polar form as:

$$q = |q|e^{\mu\theta}. \tag{6}$$

Unit pure quaternion $\mu$ and real angle $\theta$ are called the eigenaxis and eigenangle (or simply axis and angle or phase) of the quaternion (Alexiadis & Daras, 2014). The eigenaxis and eigenangle can be computed as: $\mu = V(q)/|V(q)|, \theta = \tan^{-1}(|V(q)|/S(q))$. For pure $q$, hence $S(q) = 0$, we have $\theta = \pi/2$.

In terms of number of independent parameters, or "degrees of freedom" ($DOF$), note that the magnitude $|q|$ corresponds to $1DOF$, axis $\mu$ to $2DOFs$ and $\theta$ to $1DOF$, summing to a total of $4DOF$. Quaternion multiplication is in general non-commutative, with:

$$\boldsymbol{ij} = -\boldsymbol{ji} = \boldsymbol{k}, \boldsymbol{jk} = -\boldsymbol{kj} = \boldsymbol{i}, \boldsymbol{ki} = -\boldsymbol{ik} = \boldsymbol{j}. \tag{7}$$

Note the analogy of the above formulae to vector products of 3d standard basis vectors. Indeed, these are generalized with the formula for the product of two generic quaternions $p, q \in \mathbb{H}$:

$$pq = S(p)S(q) - V(p) \cdot V(q) + S(p)V(q) + S(q)V(p) + V(p) \times V(q), \tag{8}$$

where $\cdot$ and $\times$ denote the dot and cross product respectively.

*Quaternionic convolution*: Similarly to quaternion multiplication, quaternionic convolution is non-commutative ($f * g \neq g * f$ in general, where $f, g \in \mathbb{H}^N$) and conjugation inverses the order of convolution operators ($\overline{f * g} = \overline{g} * \overline{f}$). We have left convolution and right convolution:

$$(h_L * f)[n] = \sum_{n=0}^{N-1} h_L[i]f[n-i], \tag{9}$$

$$(f * h_R)[n] = \sum_{n=0}^{N-1} f[n-i]h_R[i], \tag{10}$$

where the difference between the two formulae is whether the convolution kernel elements multiply the input from the left or right. A bi-convolution operator can also be defined (please refer to the Appendix for more details).

*Quaternion matrices, eigenvalues and eigenvectors*: Matrices with quaternion-valued entries will be denoted as $A \in \mathbb{H}^{M \times N}$ (see Fig. 1 for a visual reference). An important complication of standard matrix calculus comes with matrix eigenstructure and determinants. Due to multiplication non-commutativity, we now have two distinct ways to define eigenvalues and eigenvectors. In particular, solutions to $Ax = \lambda x$ are *left* eigenvalues and eigenvectors, while solutions to $Ax = x\lambda$ are *right* eigenvalues and eigenvectors. The sets of left and right eigenvalues are referred to as left and right spectrum, denoted $\sigma_l(\cdot)$ and $\sigma_r(\cdot)$ respectively. The two sets are in general different, with differing properties as well. Both sets can have infinite members for finite matrices, unlike real and complex matrices (cf. Fig. 2). We note here the following important lemmas on the left and right spectrum (Huang & So, 2001); we have

$$\sigma_l(pI + qA) = \{p + qt : t \in \sigma_l(A)\},$$

where $A \in \mathbb{H}^{N \times N}$, $p, q \in \mathbb{H}$, $I$ is the identity matrix in $\mathbb{H}^{N \times N}$. This property does not hold for the right spectrum.

For the right spectrum, we have

$$\sigma_r(U^H A U) = \sigma_r(A),$$

where $A, U \in \mathbb{H}^{N \times N}$ and $U$ is unitary. Also, $\lambda \in \sigma_r(A) \implies q^{-1}\lambda q \in \sigma_r(A)$ for $q \neq 0$. These properties do not hold for the left spectrum.

For additional important results, including the circular convolution, the extension of the convolution theorem, the Quaternion SVD and the Cayley-Dickson form, we refer the reader to the Appendix. Other useful results on Quaternion matrices can be found in the important treatise of Zhang (1997).

## 2.2 Discussion on difficulties and benefits associated with Quaternionic representations

Major difficulties in handling quaternionic matrix forms for the FT or the convolution operator involve the following points: a) *Multiplication non-commutativity*: Quaternion multiplication is non-commutative, a property which is passed on to quaternion matrix multiplication. b) *Multiple definitions of convolution and QFTs*. There is a left-side, a right-side, or even a two-sided Fourier transform for quaternions, as well as variations stemming from choices of different FT axes. (An analogous picture holds for quaternion convolution). c) *Difficulties with a convolution theorem*: A long discussion exists on possible adaptations of the convolution theorem to the quaternion domain (Ell & Sangwine, 2007; Cheng & Kou, 2019; Bahri et al., 2013; Pei et al., 2001; Ell et al., 2014). The convolution theorem cannot be readily applied for *any* version of the QFT and *any* version of Quaternion Convolution (Cheng & Kou, 2019). Quaternion convolution theorems

have however been proposed in the literature: Ell & Sangwine (2007) discuss a convolution theorem for the discrete QFT; Bahri et al. (2013) present results for the two-sided continuous 2D QFT. A version with the commutative bicomplex product operator also exists, holding specifically for complex signals transformed w.r.t. to axis $\boldsymbol{j}$ (Ell et al., 2014). d) *More complex eigenstructure*: Quaternion matrices have a significantly more complex eigenstructure than real or complex matrices. Due to non-commutativity of multiplication, left and right eigenvalues are defined, each corresponding to either the problem $Ax = \lambda x$ or $Ax = x\lambda$ respectively. Even worse, the number of the eigenvalues of a quaternionic matrix is in general infinite (Zhang, 1997). e) *Quaternionic determinants*: The issue of defining a Quaternion determinant is non-trivial; a function possessing all the standard properties of the complex-valued determinant cannot exist in the quaternion domain (Dyson, 1972).

But what can we gain from a matrix form and the properties of the constituent quaternionic matrices, especially when moving to the quaternion domain creates all sorts of difficulties? In a nutshell, the motivation is that vectorial signals such as color images or arrangements of rotation representations can be treated in a holistic manner, and our analysis opens up the potential to build more powerful models directly in the quaternionic domain. Given a signal processing observation model, when a Fourier operator of a Convolutional operator is defined, a matrix form allows us to easily adapt the model to the quaternion domain. This bears at least two advantages: a) The model can be easily redefined by replacing operators with their quaternion versions. b) Perhaps more importantly, solution of the model (i.e., find some parameter vector $\theta$ that fits best to observations) involves directly using properties of the Fourier and Circulant matrices (e.g. using results about eigenvalues of a sum of circulant matrices).

Properties of the related non-quaternion operations are used in state-of-the-art signal, image processing/vision and learning methods, exploiting the relation of Convolution and Fourier Transform in the real and complex domains. As examples of use, we can mention models in deblurring (Nan et al., 2020) and deconvolution (Hidalgo-Gavira et al., 2019), manipulating convolution layers of neural networks (Sedghi et al., 2018; Singla & Feizi, 2020) or constructing invertible convolutions for normalizing flows (Karami et al., 2019). These properties are well-known for the real and complex domain (Jain, 1989); *with the current work, these properties are extended and adapted from their well-known complex domain version (ℂ) to the quaternionic domain (ℍ), allowing construction of more powerful and expressive models.*

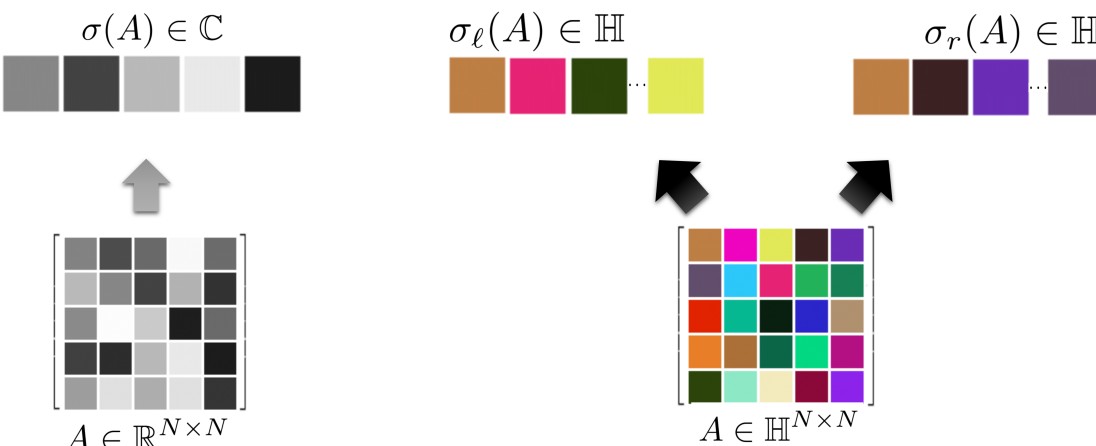

Figure 2: *The whole is more than the sum of its parts:* Quaternionic matrix eigenstructure is significantly richer than the eigenstructure of real matrices. The set of all eigenvalues of a real-valued matrix forms the spectrum $\sigma(\cdot)$ of the matrix; for real-valued (and complex-valued) matrices, it is a subset of ℂ. In quaternionic matrices, we have two distinct spectra, the left spectrum $\sigma_\ell(\cdot)$ and the right spectrum $\sigma_r(\cdot)$, corresponding to forms $Ax = \lambda x$ and $Ax = x\lambda$ respectively. No known well-posed connection exists that relates the two spectra, and in general they can both be infinite sets. The literature focuses on right eigenvalues, as no general method exists to compute the left spectrum. In this paper, our focus turns to the usefulness of the left spectrum, which we show how to compute efficiently in the case of circulant-structured matrices.

# 3 Matrix forms: Quaternion Circulant Matrix, Quaternion Fourier Matrix and their connection

The main theme of this section is exploring whether and to what extent well-known properties and relation between circulant matrices, the Fourier matrix, and their eigenstructure generalize to the quaternionic domain. The core of these properties (see for example Jain (1989) for a complete treatise) is summarized in expressing the convolution theorem in matrix form, as:

$$g = Cf \implies Ag = AA^{-1}\Lambda_C Af \implies G = \Lambda_C F \tag{11}$$

where $g, f \in \mathbb{C}^N$, $C \in \mathbb{C}^{N \times N}$ is a circulant matrix, $G, F \in \mathbb{C}^N$ are the Fourier transforms of $g, f$. As the columns of the inverse Fourier matrix $A^{-1}$ are eigenvectors of any circulant matrix, the convolution expressed by the circulant $C$ multiplied by the signal $f$ is easily written in eq. 11 as a point-wise product of the transform of the convolution kernel $\Lambda_C$ (represented as a diagonal matrix) and the transform $F$. These formulae in practice aid in handling circulant matrices, which when used in the context of signal and image processing models represent convolutional filters. The most important of these operations are with respect to compositions of circulant matrices, as well as operations in the frequency domain.

## 3.1 Quaternionic Circulant Matrices

Let $C \in \mathbb{H}^{N \times N}$ be a quaternionic circulant matrix. This is defined in terms of a quaternionic "kernel" vector, denoted as $k_C \in \mathbb{H}^N$, with quaternionic values $k_C = [c_0 \ c_1 \cdots c_{N-1}]^T$. Then, the element at row $i = 0, 1, \ldots, N-1$ and column $j = 0, 1, \ldots, N-1$ is equal to $k_C[i-j]_N$, where we take the modulo-$N$ of the index in brackets. Hence, a quaternionic circulant matrix bears the following form:

$$C^\top = \begin{pmatrix} c_0 & c_1 & c_2 & \cdots & c_{N-1} \\ c_{N-1} & c_0 & c_1 & \cdots & c_{N-2} \\ \vdots & \vdots & \vdots & \ddots & \vdots \\ c_1 & c_2 & c_3 & \cdots & c_0 \end{pmatrix}. \tag{12}$$

For any quaternionic circulant $C$, from the definition of the quaternionic circulant (eq. 12) and quaternionic convolution, we immediately have:

**Proposition 3.1.** *The product $Cx$ implements the quaternionic circular* left *convolution $k_C \circledast x$:*

$$(k_C \circledast x)[i] = \sum_{n=0}^{N-1} k_C[i-n]_N x[n], \tag{13}$$

*taken for $\forall i \in [0, N-1]$, where $x \in \mathbb{H}^N$ is the signal to be convolved, and $[\cdot]_N$ denotes modulo-$N$ indexing (Jain, 1989) (i.e., the index "wraps around" with a period equal to $N$).*

**Proposition 3.2.** *Quaternionic circulant matrices can be written as matrix polynomials:*

$$C = c_0 I + c_1 \tilde{P} + c_2 \tilde{P}^2 + \cdots + c_{N-1}\tilde{P}^{N-1}, \tag{14}$$

*where kernel $[c_0 c_1 \cdots c_N] \in \mathbb{H}^N$ and $\tilde{P} \in \mathbb{R}^{N \times N}$ is the real permutation matrix (Strang, 2019) that permutes columns $0 \to 1, 1 \to 2, \cdots, N-1 \to 0$. The inverse is also straightforward: any such matrix polynomial is also quaternionic circulant.*

**Corollary 3.2.1.** *Transpose $C^\top$ and conjugate transpose $C^H$ are also quaternionic circulant matrices, with kernels equal to $[c_0 \ c_{N-1} \cdots c_2 \ c_1]^T$ and $[\overline{c_0} \ \overline{c_{N-1}} \cdots \overline{c_2} \ \overline{c_1}]^T$ respectively.*

*Proof.* We use $\tilde{P}^H = \tilde{P}^\top = \tilde{P}^{-1}$, and $\tilde{P}^{N-a}\tilde{P}^a = I$ for any $a = 0, 1, \ldots, N$. We then have, by using Proposition 3.2:

$$C^\top = c_0 \tilde{P}^N + c_1 \tilde{P}^{N-1} + c_2 \tilde{P}^{N-2} + \cdots + c_{N-1}\tilde{P}, \tag{15}$$

and

$$C^H = \overline{c_0}\tilde{P}^N + \overline{c_1}\tilde{P}^{N-1} + \overline{c_2}\tilde{P}^{N-2} + \cdots + \overline{c_{N-1}}\tilde{P}, \tag{16}$$

hence both are polynomials over $\tilde{P}$, thus quaternionic circulant. $\square$

## 3.2 Quaternionic Fourier Matrices

We define a class of matrices as Quaternionic Fourier matrices, shorthanded as $Q_N^\mu$ for some pure unit quaternion $\mu$ (termed the "axis" of the transform) and $N \in \mathbb{N}$, as follows. The element at row $i = 0, 1, \dots, N-1$ and column $j = 0, 1, \dots, N-1$ is equal to $w_{N\mu}^{i \cdot j}$, where we have used $w_{N\mu} = e^{-\mu 2\pi N^{-1}}$, raised to the power of the product of $i, j$. Hence, we write:

$$Q_N^\mu = \frac{1}{\sqrt{N}} \begin{pmatrix} w_{N\mu}^{0 \cdot 0} & w_{N\mu}^{1 \cdot 0} & \cdots & w_{N\mu}^{(N-1) \cdot 0} \\ w_{N\mu}^{0 \cdot 1} & w_{N\mu}^{1 \cdot 1} & \cdots & w_{N\mu}^{(N-1) \cdot 1} \\ \vdots & \vdots & \ddots & \vdots \\ w_{N\mu}^{0 \cdot (N-1)} & w_{N\mu}^{1 \cdot (N-1)} & \cdots & w_{N\mu}^{(N-1) \cdot (N-1)} \end{pmatrix}. \tag{17}$$

**Proposition 3.3** (General properties). *For any Quaternionic Fourier matrix we have the following straightforward properties:*

*(1) $Q_N^\mu$ is square, Vandermonde and symmetric.*

*(2) $[Q_N^\mu]^H Q_N^\mu = Q_N^\mu [Q_N^\mu]^H = I$, $Q_N^\mu$ is unitary.*

*(3) The product $Q_N^\mu x$, where $x \in \mathbb{H}^N$, equals the* left *QFT $\mathcal{F}_L^\mu\{x\}$:*

$$F_L^\mu[u] = \frac{1}{\sqrt{N}} \sum_{n=0}^{N-1} e^{-\mu 2\pi N^{-1} n u} f[n]. \tag{18}$$

*(4) The product $\overline{Q}_N^\mu x = Q_N^{-\mu} x$, where $x \in \mathbb{H}^N$, equals the* left *inverse QFT $\mathcal{F}_L^{-\mu}\{x\}$ .*

$$F_L^{-\mu}[u] = \frac{1}{\sqrt{N}} \sum_{n=0}^{N-1} e^{+\mu 2\pi N^{-1} n u} f[n]. \tag{19}$$

*(5) $Q_N^i = A_N$ where $A_N$ is the (standard, non-quaternionic) Fourier matrix of size $N$ (Jain, 1989; Strang, 2019). The standard DFT comes as a special case of the QFT, for axis $\mu = i$ and a complex-valued input $x \in \mathbb{C}^N$.*

*(6) $Q_N^{-\mu} = (Q_N^\mu)^{-1} = \overline{Q}_N^\mu = (Q_N^\mu)^H$.*

*(7) $Q_N^\mu Q_N^\mu = \check{P}$, where $\check{P}$ is a permutation matrix that maps column $n$ to $[N - n]_N$.*

(see also the Appendix for a brief discussion over variants of the Quaternion Fourier Transform).

*Proof.* These properties are a straightforward generalization from $\mathbb{C}$, and can be confirmed by using the matrix form definition of eq. 17. Let us only add a short comment on the derivation of 3.3.7. We practically need to prove that each column of $Q_N^\mu$ is perpendicular to exactly one other column of the same matrix, and that this mapping is given by $n \to N - n$ (where we use a zero/modulo-$N$ convention, cf. Appendix B). For columns $\alpha_u$ and $\alpha_v$, we compute the result of the required product at position $(u, v)$ as:

$$\alpha_v^\top \alpha_v = \sum_{n=0}^{N-1} e^{-\mu 2\pi N^{-1} n u} e^{-\mu 2\pi N^{-1} n v}$$

$$= \sum_{n=0}^{N-1} e^{-\mu 2\pi N^{-1} n u} e^{-\mu 2\pi N^{-1} n v} e^{+\mu 2\pi N^{-1} N u},$$

which holds since $e^{\mu 2\pi u} = 1$. This is subsequently equal to

$$\sum_{n=0}^{N-1} e^{+\mu 2\pi N^{-1} n(N-u)} e^{-\mu 2\pi N^{-1} nv} = \alpha_{N-u}^H \alpha_v,$$

which is the inner product of columns $v$ and $N-u$. This will result in zero for all pairs of $v, N-u$ except for when $v = [N-u]_N$. Hence, the required product will have all-zero columns save for exactly one element, different for each column; this is by definition a permutation matrix (Strang, 2019). $\square$

Consequently, and in contrast to what holds in the complex domain, we have an infinite number of different Fourier matrices for a given signal length $N$, one for each different choice of pure unit axis $\mu$. Proposition 3.3 stated that by flipping the sign of the axis we obtain the inverse QFT with respect to the same axis. In general, two arbitrary Fourier matrices are connected via a rotation of their components:

**Proposition 3.4.** *Let $Q_N^\mu$, $Q_N^\nu$ Quaternionic Fourier matrices with non-collinear axes $\mu, \nu$. We can always find unit $p \in \mathbb{H}$ such that*

$$Q_N^\mu = pQ_N^\nu \bar{p} \tag{20}$$

*The required quaternion $p$ is $e^{\xi\theta/2}$, where $\xi = \nu\mu + V(\nu) \cdot V(\mu)$ and $\theta = arccos(V(\mu) \cdot V(\nu))$.*

*Proof.* For any $w_{N\mu}$ and $w_{N\nu}$, we can find $p \in \mathbb{H}$ such that $w_{N\mu} = pw_{N\nu}p^{-1}$. Since any pure unit quaternions $\nu, \mu$ are situated on the unit sphere $\{t \in \mathbb{R}i + \mathbb{R}j + \mathbb{R}k : |t| = 1\}$, we can obtain the one from the other by applying a rotation about the origin. Consequently, there exists $p \in \mathbb{H}$ so that $\mu = p\nu p^{-1}$ (Stillwell, 2008) (note that $w_{N\mu}, w_{N\nu}$ are unit but not necessarily pure). From there, assuming pure unit $\mu$ and real $\theta$ we have the following steps: $e^{\mu\theta} = e^{p\nu p^{-1}\theta} = pp^{-1}cos\theta + p\nu p^{-1}sin\theta = p(cos\theta + \nu sin\theta)p^{-1} = pe^{\nu\theta}p^{-1}$.

It suffices to plug $\theta = 2\pi N^{-1}$ in the previous equation to conclude the required $w_{N\mu} = pw_{N\nu}p^{-1}$. The same transform can be used for all powers of $w_{N\mu}$, $w_{N\nu}$ to any exponent $\gamma$, and indeed we have $pw_{N\mu}^\gamma p^{-1}$ $= pw_{N\mu}p^{-1}pw_{N\mu}p^{-1}p \cdots w_{N\mu}p^{-1} = w_{N\nu}w_{N\nu} \cdots w_{N\nu} = w_{N\nu}^\gamma$. Since any element of $Q_N^\mu$ can be written as a power $w_{N\mu}^\gamma$ for some natural exponent $\gamma$, we can write $Q_N^\mu = pQ_N^\nu p^{-1}$, which applies the same transform on all elements simultaneously. Intuitively, this action represents a rotation of the unitary disc where all elements of $Q_N^\mu$ are situated.

The axis of rotation is given by the cross product of the two pure unit vectors $\mu, \nu$, as it is by definition perpendicular to the plane that $\mu, \nu$ form. The product $V(\nu) \times V(\mu)$ is equal to $\nu\mu + V(\nu) \cdot V(\mu)$ as $S(\mu) = S(\nu) = 0$, hence we compute $\xi = \nu\mu + V(\nu) \cdot V(\mu)$. Also, $V(\mu) \cdot V(\nu) = |\mu||\nu|cos(\theta) \implies \theta = arccos(V(\mu) \cdot V(\nu))$. $\square$

Let us add a short comment on the intuition behind this relation. We must note that all elements of $Q_N^\mu$ are situated on the same plane in $\mathbb{H}$, which is different from the plane for elements of $Q_N^\nu$ (due to the assumption of non-collinearity). However, for any given axis, all planes intersect the origin and all pass from $(1, 0, 0, 0) \in \mathbb{H}$, since element $[Q_N^\mu]_{1,1}$ equals 1 for any $\mu$ and $N \in \mathbb{N}$. Elements of $Q_N^\mu$ are situated on a unit circle on their corresponding plane, whereupon their position is defined by their relative angle to the line passing between the origin and $(1, 0, 0, 0)$. This position is determined by the value of the element $w_{N\mu}$, and the power to which this element is raised, so as to give the element of $Q_N^\mu$ in question. Intuitively, this action represents a rotation of the unitary disc where all elements of $Q_N^\mu$ are situated (see Fig. 3 for a visual reference of this concept).

## 3.3 Connection of Circulant and Fourier matrices

The following results can then be proved, underpinning the relation between Quaternionic Circulant and Quaternionic Fourier matrices, in particular with respect to the left spectrum of the former:

**Proposition 3.5** (Circulant & Fourier Matrices). *For any $C \in \mathbb{H}^{N \times N}$ that is circulant, and any pure unit $\mu \in \mathbb{H}$,*

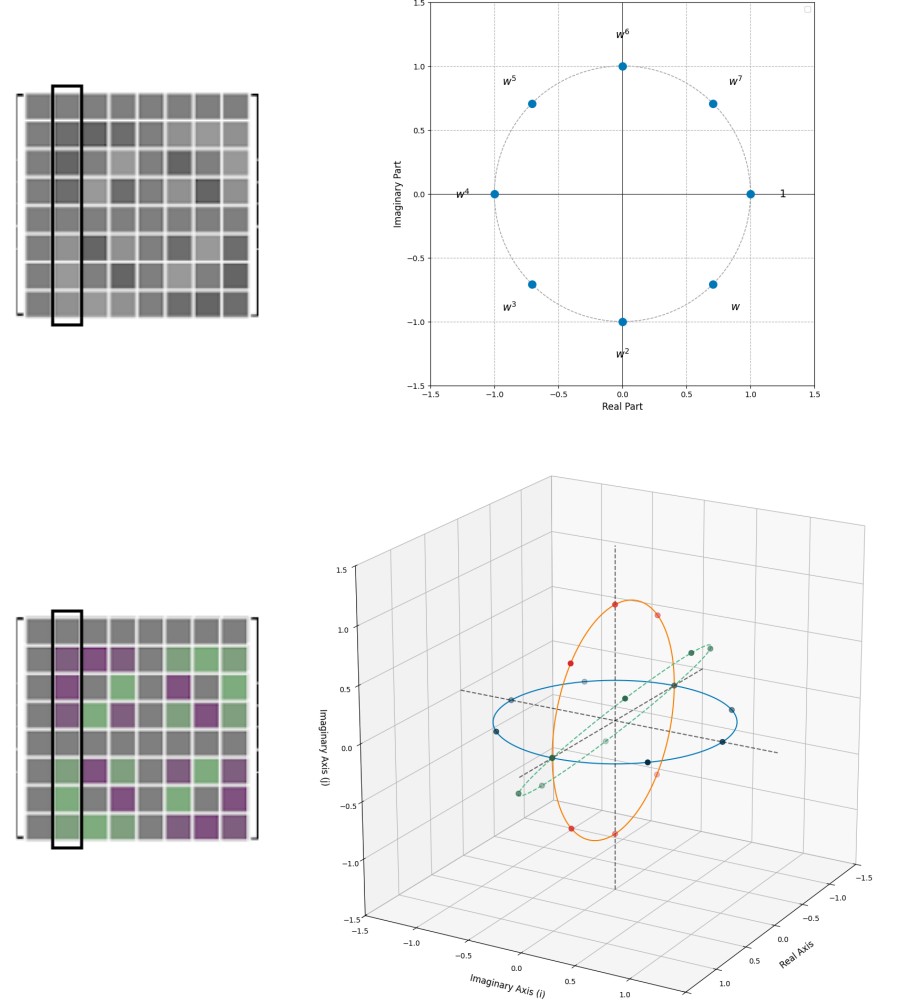

Figure 3: The relation of the DFT matrix to the QFT matrix. In the top row, we see a visualization of a $8 \times 8$ Discrete Fourier Transform (DFT) matrix (top left) and a plot of the elements of its *second column*. All elements are powers of $w = e^{-2i\pi/8}$, and they elegantly form a unitary disc one the real-complex plane (top right). In the bottom row, we see a visualization of a $8 \times 8$ Quaternionic Fourier Transform (DFT) matrix (bottom left), and a plot of the same column. Here, elements are powers of $w = e^{-2\mu\pi/8}$, where unitary $\mu \in \mathbb{H}$ is called the axis of the transform (bottom right). Similar considerations are in place for the rest of the columns in both matrices, as all elements are powers of $w$. Again, elements form a disc that lays on a plane, the orientation of which is controlled by $\mu$. For $\mu = i$, we obtain the DFT matrix. (For purposes of visualization: a) The top-left plot represents the DFT matrix which is complex ($\mathbb{C}$), but is depicted in grayscale in order to emphasize its relation with the QFT matrix as a special case for $\mu = i$; b) The bottom-right plot shows a projection of $\mathbb{H}$ to space $k = 0$.)

(1) *Any column $k = 0, 1, \ldots, N-1$ of the inverse QFT matrix $Q_N^{-\mu}$ is an eigenvector of $C$. Column $k$ corresponds to the $k^{th}$ component of the vector of* left *eigenvalues $\boldsymbol{\lambda}^\mu = [\lambda_1^\mu \lambda_2^\mu \cdots \lambda_N^\mu]^T \in \mathbb{H}^N$. Vector $\boldsymbol{\lambda}^\mu$ is equal to the* right *QFT $\mathcal{F}_{R*}^\mu$ of the kernel of $C$.*

(2) *Any column $k = 0, 1, \ldots, N-1$ of the inverse QFT matrix $Q_N^{-\mu}$ is an eigenvector of $C^H$. Column $k$ corresponds to the $k^{th}$ component of the vector of* left *eigenvalues $\boldsymbol{\kappa}^\mu = [\kappa_1^\mu \kappa_2^\mu \cdots \kappa_N^\mu]^T \in \mathbb{H}^N$. The conjugate of the vector $\boldsymbol{\kappa}^\mu$ is equal to the* left *QFT $\mathcal{F}_{L*}^\mu$ of the kernel of $C$.*

where transforms denoted with an asterisk (*) refer to using a unitary coefficient instead of $1/\sqrt{N}$:

$$F_{L*}^{\mu}[u] = \sum_{n=0}^{N-1} e^{-\mu 2\pi N^{-1} nu} f[n], \quad F_{R*}^{\mu}[u] = \sum_{n=0}^{N-1} f[n] e^{-\mu 2\pi N^{-1} nu}. \tag{21}$$

Note here that the left spectrum becomes important concerning the circulant matrix eigenstructure. (In general, quaternionic matrices have two different spectra, corresponding to left and right eigenvalues. For real matrices, the two spectra will coincide, however the exact way these two sets are connected is largely unknown in the case of true quaternionic matrices (Zhang, 1997; Macías-Virgós et al., 2022)).

*Proof.* Let $\alpha_k$ the $k^{th}$ column of $Q_N^{-\mu}$. Its product with quaternionic circulant $C$ is computed as:

$$\sqrt{N} C \alpha_k = \begin{pmatrix} \sum_{n=0}^{N-1} h[0-n]_N w_{N\mu}^{-kn} \\ \sum_{n=0}^{N-1} h[1-n]_N w_{N\mu}^{-kn} \\ \sum_{n=0}^{N-1} h[2-n]_N w_{N\mu}^{-kn} \\ \vdots \\ \sum_{n=0}^{N-1} h[N-1-n]_N w_{N\mu}^{-kn} \end{pmatrix}, \tag{22}$$

where $h[i]_N$ is the $i^{th}$/modulo-$N$ element of $h$; $h$ is assumed to be the kernel of circulant matrix $C$. The $m^{th}$ element of $C\alpha_k$ is then:

$$= \frac{1}{\sqrt{N}} \sum_{n=0}^{N-1} h[m-n]_N w_{N\mu}^{-kn} = \frac{1}{\sqrt{N}} \sum_{\ell=m}^{m-(N-1)} h[\ell]_N w_{N\mu}^{-kn}$$

$$= \frac{1}{\sqrt{N}} [\sum_{\ell=m}^{m-(N-1)} h[\ell]_N w_{N\mu}^{k\ell}] w_{N\mu}^{-km} = \sum_{\ell=0}^{N-1} [h[\ell]_N w_{N\mu}^{k\ell}] \frac{1}{\sqrt{N}} w_{N\mu}^{-km}, \tag{23}$$

where we used $\ell = m - n$ and in the last form of equation 23, the term inside the brackets ($\sum_{\ell=0}^{N-1} h[\ell]_N w_{N\mu}^{k\ell}$) we have the $k^{th}$ element of the right-side QFT $\mathcal{F}_{R*}^{\mu}$ of $h$. (Note that this is the *asymmetric* version of the QFT, cf. Appendix). Outside the brackets, $\frac{1}{\sqrt{N}} w_{N\mu}^{-km}$ can be identified as the $m^{th}$ element of $\alpha_k$, shorthanded as $[\alpha_k]_m$. Therefore, $C\alpha_k = \lambda \alpha_k$, so $\alpha_k$ is an eigenvector of $C$; by the previous argument, $\lambda$ is equal to the $k^{th}$ element of the right-side QFT. The second part of the theorem, concerning $C^H$, is dual to the first part. $\qquad \square$

**Corollary 3.5.1.** *For any pure unit axis $\mu \in \mathbb{H}$, the conjugates of the eigenvalues $\boldsymbol{\lambda}^{\mu}$ and $\boldsymbol{\kappa}^{\mu}$ are also left eigenvalues of $C^H$ and $C$ respectively.*

*Proof.* We will prove the corrolary for $\overline{\lambda^{\mu}}$ and $C^H$ as the case of $\overline{\kappa^{\mu}}$ and $C$ is dual to the former one. We have $Cw = \lambda_i^{\mu} w \implies (C - \lambda_i^{\mu} I)w = 0$ given non-zero eigenvector $w$, hence the nullspace of $C - \lambda_i^{\mu} I$ is non-trivial. From this, we have $\text{rank}[C - \lambda_i^{\mu} I] < N \implies \text{rank}[(C - \lambda_i^{\mu} I)]^H < N$ and $\text{rank}[C^H - \overline{\lambda_i^{\mu}}] < N$, where we have used the property $\text{rank}(A) = \text{rank}(A^H)$ (Zhang, 1997, theorem 7.3). Thus, the nullspace of $C^H - \overline{\lambda_i^{\mu}} I$ is nontrivial, and $\overline{\lambda_i^{\mu}}$ is an eigenvalue (note that this proof does not link these eigenvalues to specific eigenvectors, however). $\qquad \square$

**Corollary 3.5.2.** *For any pure unit axis $\mu \in \mathbb{H}$, the vector of left eigenvalues $\boldsymbol{\lambda}^{\mu}$ is a flipped version of $\boldsymbol{\lambda}^{-\mu}$, where the DC component and the $(N/2)^{th}$ component (zero-indexed) remain in place.*

*Proof.* We want to prove that the $i^{th}$ component of $\boldsymbol{\lambda}^{\mu}$ ($\lambda_i^{\mu}$) is equal to the $[N-i]_N^{th}$ component of $\boldsymbol{\lambda}^{-\mu}$ ($\lambda_{N-i}^{\mu}$, where ordering is w.r.t. modulo-$N$ indexing). But the $\lambda_i^{\mu}$ is a left eigenvalue that corresponds to the $i^{th}$ column of the QFT matrix with axis $\mu$ as its eigenvector, and also equal to the $i^{th}$ component of the

right QFT by Proposition 3.5. It is equal to $\sum_{n=0}^{N-1} f[n]e^{-\mu 2\pi N^{-1}ni}$, while the $[N-i]_N^{th}$ component of $\lambda^{-\mu}$ is equal to $\sum_{n=0}^{N-1} f[n]e^{+\mu 2\pi N^{-1}n(N-i)}$ by definition. Note however that:

$$\sum_{n=0}^{N-1} f[n]e^{-\mu 2\pi N^{-1}ni} = \sum_{n=0}^{N-1} f[n]e^{-\mu 2\pi N^{-1}ni}e^{+\mu 2\pi N^{-1}Nn} = \sum_{n=0}^{N-1} f[n]e^{+\mu 2\pi N^{-1}n(N-i)},$$

because $e^{\mu 2\pi n} = 1$ for any $n \in \mathbb{Z}$ and pure unit $\mu$. Hence $\lambda_i^{\mu} = \lambda_{[N-i]_N}^{-\mu}$. The only components for which $i = [N-i]_N$ are $i = 0$ (DC component) and $i = N/2$, hence for these we have $\lambda_i^{\mu} = \lambda_i^{-\mu}$. □

**Proposition 3.6.** *A set of $N$ left eigenvalues that correspond to the eigenvectors - columns of $Q_N^{-\mu}$ for some choice of axis $\mu \in \mathbb{H}$, uniquely defines a circulant matrix $C$. The kernel of $C$ is computed by taking the inverse right QFT of the vector of these $N$ left eigenvalues.*

Note that while the analogous proposition is known to hold true for non-quaternionic matrices and the inverse Fourier matrix, it is not completely straightforward to show for quaternionic matrices. For complex matrices it can be proven true by using e.g. the spectral theorem (Strang, 2019), however no such or analogous proposition is known concerning quaternionic matrices and their *left* spectrum.

*Proof.* We shall proceed by proving the required proposition by contradiction. Let $C \neq D \in \mathbb{H}^{N \times N}$ be circulant matrices; all columns of $Q_N^{-\mu}$ for a choice of axis $\mu$ are eigenvectors of both $C$ and $D$ due to proposition 3.5. We then have:

$$[C\alpha_k]_m = [\sum_{\ell=0}^{N-1} h[\ell]_N w_{N\mu}^{k\ell}]\frac{1}{\sqrt{N}}w_{N\mu}^{-km},$$

$$[D\alpha_k]_m = [\sum_{\ell=0}^{N-1} g[\ell]_N w_{N\mu}^{k\ell}]\frac{1}{\sqrt{N}}w_{N\mu}^{-km},$$

for all $m \in [0, N-1]$, where $h, g$ are the convolution kernels of $C, D$ respectively, and $\alpha_k$ corresponds to the $k^{th}$ column of $Q_N^{-\mu}$. By the assumption we have set in this proof, we have that left eigenvalues are equal for corresponding eigenvectors for either matrix, hence:

$$\sum_{\ell=0}^{N-1} h[\ell]_N w_{N\mu}^{k\ell} = \sum_{\ell=0}^{N-1} g[\ell]_N w_{N\mu}^{k\ell},$$

or, written in a more compact form,

$$\mathcal{F}_{R*}\{h\} = \mathcal{F}_{R*}\{g\}.$$

However due to the invertibility of the (either left or right-side) QFT, we have $h = g \implies C = D$, which contradicts the assumption. Hence, circulant $C$ must be unique. □

Note that an implication of this proposition is that, while the number of left eigenvalues is in general infinite, picking a random set of $N$ left eigenvalues does not suffice to reconstruct the original circulant matrix. It must be ensured that the selected left eigenvalues correpond to columns of $Q_N^{-\mu}$, where we are free to choose the axis $\mu \in \mathbb{H}$.

**Corollary 3.6.1.** *Given (a vector, or ordered set of) $N$ left eigenvalues $\boldsymbol{\lambda}^{\mu}$, we can use the QFT to reconstruct a circulant matrix $C \in \mathbb{H}^{N \times N}$ with $\boldsymbol{\lambda}^{\mu} \subset \sigma_l(C)$. The corresponding eigenvectors are the columns of the QFT matrix $Q_N^{-\mu}$. The resulting matrix $C$ is unique, in the sense that it is the only matrix $\in \mathbb{H}^{N \times N}$ with this pair of left eigenvalues and eigenvectors.*

*Proof.* This follows from Propositions 3.5 and 3.6: From Proposition 3.5 we have the result that $Q_N^{-\mu}$ contains eigenvectors for any circulant matrix $C \in \mathbb{H}^{N \times N}$. This allows us to reconstruct the matrix by applying the Quaternionic Fourier Transform over the given vector of left eigenvalues $\boldsymbol{\lambda}^{\mu}$. From Proposition 3.6, we know that there is no other quaternionic matrix with the same left eigenvalues $\boldsymbol{\lambda}^{\mu}$, given axis $\mu$ for our eigenvectors (i.e. eigenvectors chosen as columns of $Q_N^{-\mu}$). □

**Corollary 3.6.2.** *Writing proposition 3.5 in a matrix diagonalization form ($A = S\Lambda S^{-1}$), where the columns of $S$ are eigenvectors and $\Lambda$ is a diagonal matrix of* left *eigenvalues, is not possible. This would require us to be able to write $CQ_N^{-\mu} = Q_N^{-\mu}\Lambda^\mu$; however, the right side of this equation computes right eigenvalues, while proposition 3.5 concerns left eigenvalues.*

*Proof.* The given diagonalization implies that we can rearrange terms to write $AS = S\Lambda$. However, by definition of quaternionic matrix multiplication this form corresponds to $Ax = x\lambda$, which defines *right* eigenvalues (Zhang, 1997). For left eigenvalues we would require to write $Ax = \lambda x$, and $\lambda x = x\lambda$ is in general not true, due to noncommutativity of $\mathbb{H}$. Rephrased as a proof by contradiction, by the above argument any diagonalization $A = S\Lambda S^{-1}$ is related to right eigenvalues, and obtaining left eigenvalues by such a form would mean that left and right spectrums coincide. This is known to not be the case in general (Zhang, 1997). □

Consequently, and despite an only partial generalization of the corresponding well-known properties to quaternionic circulant matrices, we must note that quaternion circulant matrices have the rather singular property of being relatively easy to numerically compute (part of its) left spectrum:

**Corollary 3.6.3.** *For any Quaternionic Circulant $C$, any* right-side QFT $\mathcal{F}_{R*}^\mu$ *(i.e., with respect to arbitrary pure unit axis $\mu$) of quaternionic convolution kernel $k_C$ will result to a vector of left eigenvalues for $C$.*

*Proof.* This stems from Proposition 3.5. Given a circulant matrix $C \in \mathbb{H}^{N \times N}$, we obtain its kernel $k_C$ as its first column and apply right-side QFT (eq. 21). From Proposition 3.5, we know that the resulting vector $\boldsymbol{\lambda}^\mu$ equals to left eigenvalues of $C$. (Note that we can change $\mu$ at will, and obtain a different set of left eigenvalues). □

Concerning computation of the left spectrum, in general no procedure is known to be applicable to a generic (non-circulant) quaternionic matrix (Zhang, 1997; Macías-Virgós et al., 2022). *Only* its right spectrum can be fully calculated using a well-defined numerical procedure (Le Bihan & Sangwine, 2003).

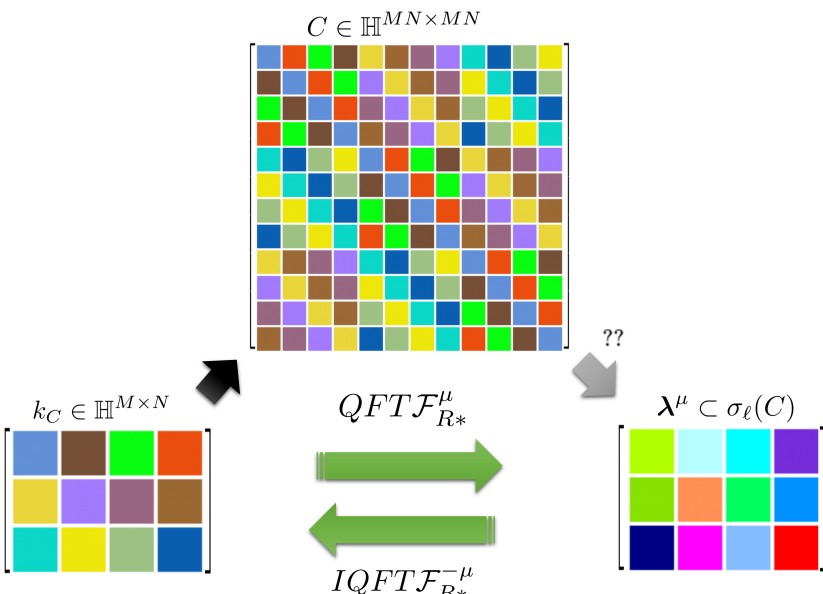

Figure 4: A synopsis of our key results in Propositions 3.5 and 3.9. Quaternion convolution kernel $k_C$ (left) is related to doubly-block circulant matrix $C$ (top). In this work, we show that the left spectrum (right) of any $C$ can be accessed via applying a QFT over kernel $k_C$. Furthermore, the kernel can be reconstructed exactly via IQFT. This provides a fast and accurate way to access spectral characteristics of convolution, completely bypassing the construction and manipulation of the (quadratically increasing) doubly-block circulant $C$.

**Proposition 3.7** (Eigenstructure of sums, products and inverse of circulant matrices)**.**

*Let $L, K \in \mathbb{H}^{N \times N}$ circulant matrices, then the following propositions hold. (Analogous results hold for $L, K$ doubly-block circulant).*

*(a) The sums $L + K$ and products $LK$, $KL$ are also circulant. Any scalar product $pL$ or $Lp$ where $p \in \mathbb{H}$ also results to a circulant matrix.*

*(b) Given some pure unit axis $\mu \in \mathbb{H}$, let $w$ be the $i^{th}$ column of the inverse QFT matrix $Q_N^{-\mu}$, and $\lambda_i^{\mu}, \kappa_i^{\mu}$ are left eigenvalues of $L, K$ with respect to the shared eigenvector $w$. Then, $w$ is an eigenvector of...*

*i. $L + K$ with left eigenvalue equal to $\lambda_i^{\mu} + \kappa_i^{\mu}$.*

*ii. $pL$ with left eigenvalue equal to $p\lambda_i^{\mu}$.*

*iii. $Lp$ with left eigenvalue equal to $\lambda_i^{p\mu p^{-1}} p$ .*

*iv. $LK$ with left eigenvalue equal to $\lambda_i^{\kappa_i^{\mu} \mu [\kappa_i^{\mu}]^{-1}} \kappa_i^{\mu}$ if $\kappa_i^{\mu} \neq 0$ and equal to $0$ if $\kappa_i^{\mu} = 0$.*

*v. Provided $L^{-1}$ exists, $\lambda_i^{\mu} w (\lambda_i^{\mu})^{-1}$ is an eigenvector of $L^{-1}$ with left eigenvalue equal to $(\lambda_i^{\mu})^{-1}$.*

*Proof.* (a) For proving $L + K$, $pL$ and $Lp$ are circulant, it suffices to use their matrix polynomial form (cf. Proposition 3.2). Then summation and scalar product are easily shown to be also matrix polynomials w.r.t. the same permutation matrix, hence also circulant.

For the product $LK$, we decompose the two matrices as linear combinations of real matrices, writing

$$LK = (L_e + L_i \boldsymbol{i} + L_j \boldsymbol{j} + L_k \boldsymbol{k})(K_e + K_i \boldsymbol{i} + K_j \boldsymbol{j} + K_k \boldsymbol{k}),$$

where $L_e, L_i, L_j, L_k, K_e, K_i, K_j, K_k \in \mathbb{R}^{N \times N}$ and circulant. Computation of this product results in 16 terms, which are all produced by multiplication of a real circulant with another real circulant, and multiplication by a quaternion imaginary or real unit. All these operations result to circulant matrices (for the proof that the product of real circulant matrices equals a circulant matrix see e.g. Jain (1989)), as does the summation of these terms. $KL$ is also circulant by symmetry, however not necessarily equal to $LK$, unlike in the real matrix case.

(b) i. $(L + K)w = Lw + Kw = \lambda_i^{\mu} w + \kappa_i^{\mu} w = (\lambda_i^{\mu} + \kappa_i^{\mu})w$. Hence $w$ is an eigenvector of $L + K$ with left eigenvalue equal to $\lambda_i^{\mu} + \kappa_i^{\mu}$.

ii. $(pL)w = p(Lw) = (p\lambda_i^{\mu})w$. Hence $w$ is an eigenvector of $pL$ with left eigenvalue equal to $p\lambda_i^{\mu}$.

iii. Let $p_u$ a unit quaternion with $p_u = p/|p|$, assuming $p \neq 0$.

$$(Lp)w = |p|(Lp_u)w = |p|Lp_u w p_u^{-1} p_u = |p|Lzp_u, \tag{24}$$

where $z = p_u w p_u^{-1}$ represents a rotation of $w$ to vector $z$ which is also an eigenvector of $L$; that is because the $k^{th}$ element of $w$ is equal to $e^{-ik\mu 2\pi N^{-1}}$. With a similar manipulation as in Proposition 3.4, $z$ is a column of the inverse QFT matrix $Q_N^{-p_u \mu p_u^{-1}}$, i.e. an inverse QFT matrix with an axis different to the original $\mu$. Thus, due to Proposition 3.5, $z$ is an eigenvector of the circulant $L$, the left eigenvalue of which is $\lambda_i^{p_u \mu p_u^{-1}}$. Continuing eq. 24, we write:

$$|p|Lzp_u = |p|\lambda_i^{p_u \mu p_u^{-1}} z p_u = |p|\lambda_i^{p_u \mu p_u^{-1}} p_u w p_u^{-1} p_u \implies$$

$$(Lp)w = (\lambda_i^{p_u \mu p_u^{-1}} p)w, \tag{25}$$

hence $w$ is an eigenvector of $Lp$ with left eigenvalue equal to $\lambda_i^{p_u \mu p_u^{-1}} p$.

iv. $LKw = L\kappa_i^{\mu} w = \lambda_i^{q_u \mu q_u^{-1}} \kappa_i^{\mu} w$, where $q = \kappa_i^{\mu}, q_u = q/|q|$ and we used the right scalar product result of (iii), supposing that $q \neq 0$. Hence $w$ is an eigenvector of $LK$ with left eigenvalue equal to $\lambda_i^{q_u \mu q_u^{-1}} \kappa_i^{\mu}$. If $q = 0$, we cannot use the result of (iii); the resulting eigenvalue is simply equal to 0, since $LKw = 0$.

v. $Lw = \lambda_i^\mu w \implies Lw(\lambda_i^\mu)^{-1} = \lambda_i^\mu w(\lambda_i^\mu)^{-1} \implies w(\lambda_i^\mu)^{-1} = L^{-1}\lambda_i^\mu w(\lambda_i^\mu)^{-1} \implies L^{-1}\lambda_i^\mu w(\lambda_i^\mu)^{-1} = (\lambda_i^\mu)^{-1}\lambda_i^\mu w(\lambda_i^\mu)^{-1} \implies L^{-1}z = (\lambda_i^\mu)^{-1}z$ where $z = \lambda_i^\mu w(\lambda_i^\mu)^{-1}$. $\qquad\square$

### 3.4 Quaternionic Doubly-Block Circulant Matrices

Block-circulant matrices are block matrices that are made up of blocks that are circulant. Doubly-block circulant matrices are block-circulant with blocks that are circulant themselves. These latter are useful in representing 2D convolution (Jain, 1989). Results for circulant matrices in general are valid also for doubly-block circulant matrices, where the role of the Quaternionic Fourier Matrix $Q_N^{-\mu}$ is taken by the Kronecker product $Q_M^{-\mu} \otimes Q_N^{-\mu}$.

**Proposition 3.8** (Doubly Block-Circulant & Fourier Matrices). *Let $D \in \mathbb{H}^{MN \times MN}$ be doubly block-circulant, with $N \times N$ blocks, and each block is sized $M \times M$. Then the product $D \cdot vec(x)$, implements the quaternionic circular* left *2D convolution $k_D \circledast x$. The operator $vec(x)$ represents column-wise vectorization of signal $x \in \mathbb{H}^{M \times N}$.*

$$(k_D \circledast x)[i,j] = \sum_{m=0}^{M-1} \sum_{n=0}^{N-1} k_D[i-m, j-n]_{M,N} x[m,n], \qquad (26)$$

*Proof.* A doubly block-circulant matrix $D$ can be written in block form as:

$$D^\top = \begin{pmatrix} D_0 & D_1 & D_2 & \cdots & D_{N-1} \\ D_{N-1} & D_0 & D_1 & \cdots & D_{N-2} \\ \vdots & \vdots & \vdots & \ddots & \vdots \\ D_1 & D_2 & D_3 & \cdots & D_0 \end{pmatrix},$$

where $\{D_\tau\}_{\tau=0}^{N-1}$ is a set of $N$ circulant blocks, and for each of the blocks we have $D_\tau \in \mathbb{H}^{M \times M}$ by definition (note the analogy to eq. 12). Block-wise, $D$ is circulant, and each block $D_\tau$ is circulant itself.

Let us denote column $v$ of $x$ as $x_v$, so $vec(x) = [x_0^\top \ x_1^\top \cdots x_{N-1}^\top]^\top$. Before considering the required product $D \cdot vec(x)$, we can observe that it should be made up of linear combinations of blocks $D_\tau$ and columns $x_v$. Since $D_\tau$ is circulant, it holds that these products represent a set of $1D$ quaternionic circular left convolutions (cf. eq. 42)

$$D_\tau x_v = \sum_{m=0}^{M-1} k_D[i-m, \tau]_M x[m,v], \quad \forall(v,\tau) \in [0, N-1]^2.$$

The required product $D \cdot vec(x)$ can then be written as:

$$\begin{pmatrix} \sum_{m=0}^{M-1} \sum_{n=0}^{N-1} k_D[i-m, 0-n]_{M,N} x[m,n] \\ \sum_{m=0}^{M-1} \sum_{n=0}^{N-1} k_D[i-m, 1-n]_{M,N} x[m,n] \\ \sum_{m=0}^{M-1} \sum_{n=0}^{N-1} k_D[i-m, 2-n]_{M,N} x[m,n] \\ \vdots \\ \sum_{m=0}^{M-1} \sum_{n=0}^{N-1} k_D[i-m, N-1-n]_{M,N} x[m,n] \end{pmatrix},$$

which coincides with the vectorized form of the quaternionic circular left 2D convolution $k_D \circledast x$. $\qquad\square$

**Proposition 3.9.** *Let $R = Q_M^{-\mu} \otimes Q_N^{-\mu}$ and the invertible coordinate mapping $k \leftrightarrow i + Mj$. For any $C \in \mathbb{H}^{MN \times MN}$ that is doubly block-circulant and any pure unit $\mu \in \mathbb{H}$,*

(1) *Any column $k = 0, 1, \ldots, (M-1)(N-1)$ of $R$ is an eigenvector of $C$. Column $k$ corresponds to the $(i,j)$ component of the matrix of* left *eigenvalues $\boldsymbol{\lambda}^\mu \in \mathbb{H}^{M \times N}$. Matrix $\boldsymbol{\lambda}^\mu$ is equal to the* right *QFT $\mathcal{F}_{R*}^\mu$ of the kernel of $C$.*

(2) *Any column $k = 0, 1, \ldots, (M-1)(N-1)$ of $R$ is an eigenvector of $C^H$. Column $k$ corresponds to the $(i,j)$ component of the matrix of* left *eigenvalues $\boldsymbol{\kappa}^\mu \in \mathbb{H}^{M \times N}$. The conjugate of the matrix $\boldsymbol{\kappa}^\mu$ is equal to the* left *QFT $\mathcal{F}_{L*}^\mu$ of the kernel of $C$.*

*Proof.* The proof for the doubly-block circulant / 2D case is analogous to the 1D case. Let $\alpha_k$ the $k^{th}$ column of $R = Q_M^{-\mu} \otimes Q_N^{-\mu}$. The $\ell^{th}$ element of $C\alpha_k$ is:

$$
[C\alpha_k]_\ell = \frac{1}{\sqrt{M}} \frac{1}{\sqrt{N}} \sum_{m=0}^{M-1} \sum_{n=0}^{N-1} h[q-m, s-n]_{M,N} w_{M\mu}^{-xm} w_{N\mu}^{-yn}
$$
$$
[\sum_{p=0}^{M-1} \sum_{r=0}^{N-1} h[p,r]_{M,N} w_{M\mu}^{xp} w_{N\mu}^{yr}] \frac{1}{\sqrt{MN}} w_{M\mu}^{-xq} w_{N\mu}^{-ys},
$$

(27)

where we used $\varpi(\ell) = (q, s)$ and $p = q - m$ and $r = s - n$. Note that $w_{M\mu}$ and $w_{N\mu}$ commute between themselves because they share the same axis $\mu$, but they do not commute with the kernel element $h$. $\square$

The reader may examine Fig. 4 as a visual summary of propositions 3.5 and 3.9.

# 4 Proof-of-Concept Application: Fast computation of singular values & Bounding of the Lipschitz Constant

We present a method to bound the Lipschitz constant of a Neural Network that comprises Quaternionic Convolutions, based on our theoretical results. Determination and bounding of the Lipschitz constant has been shown to be a way to control the generalization error (Prince, 2023, Chapter 9, Appendix B). Other important uses include application in Wasserstein GANs (Petzka et al., 2018), (Prince, 2023, Chapter 15) or plug-and-play networks (Ryu et al., 2019) and deep equilibrium (DEQ) networks (Pabbaraju et al., 2021). While the literature on Lipschitz constant bounding or other regularization methods over real-valued operations is rich, with many of them focusing on the specifics of convolutional operations (Grishina et al., 2024), the literature on methods applicable to $\mathbb{H}$ is comparatively not as extensive (Altamirano-Gomez & Gershenson, 2024).

The value of the Lipschitz constant depends on the product of the spectral norms of the weight matrices of the network. Given any linear layer, a bound $c$ on its spectral norm can be set by projecting the operation matrix onto the set of linear transforms with a bounded norm (Lefkimmiatis et al., 2013; Sedghi et al., 2018). We are interested in computing and constraining the maximum value of $\|f^\ell(x)\|$, where $f^\ell$ represents the operator in layer $\ell$ and $x$ is the layer input, for all layers in the network. For an arbitrary linear layer, expressed as matrix $B \in \mathbb{H}^{M \times N}$, this leads to dealing with a Rayleigh quotient of the form $\frac{x^H B^H B x}{x^H x}$, and from here we can use the Quaternion SVD (Sangwine & Le Bihan, 2006) to proceed. We analyze $B$ as $U, \Sigma, V^H = \text{qsvd}(B)$ and reconstruct the regularized operation as $U\check{\Sigma}V^H$, where $\check{\Sigma}$ contains singular values projected in the range $[0, c]$. Note that in $\mathbb{H}$, singular values are related to the *right* spectrum (cf. Appendix). When the linear layer in question corresponds to a Quaternionic Convolution, matrix $B$ is circulant. While we can still use the QSVD over $B$ to bound the spectrum, the required related space and time complexity is prohibitive in practice (we refer to this method as "brute-force", see further below for a comparison and discussion). We can instead circumvent computing the "expensive" doubly-block circulant matrix altogether, using our results.

We will present the approach as a two-step process. First, we will show how to compute singular values for the convolution operation. Second, we will discuss how we can reconstruct a convolution using the clipped singular values.

*Fast Computation of Singular Values.* We first require the additional lemmas that follow (please refer to the appendix for their proof):

**Lemma 4.1.** *For any pure unit $\mu \in \mathbb{H}$ and any $N \in \mathbb{N}$, the right span of the columns of $Q_N^\mu$ is $\mathbb{H}^N$. For any pure unit $\mu \in \mathbb{H}$ and any $M, N \in \mathbb{N}$, the right span of the columns of $Q_M^\mu \otimes Q_N^\mu$ is $\mathbb{H}^{MN}$.*

**Lemma 4.2.** *Let $A \in \mathbb{H}^N$ be block-diagonal. The left eigenvalues of each block of $A$ are also left eigenvalues of $A$. The right eigenvalues of each block of $A$ are also right eigenvalues of $A$.*

The first key result for computing singular values is:

**Proposition 4.3.** *Let $C \in \mathbb{H}^{N \times N}$ be quaternionic circulant. Then the square norm $\|Cx\|^2$, given some $x \in \mathbb{H}^N$, can be written as $c^H \Xi c$, where $c, x \in \mathbb{H}^N$ are related by a bijection, and $\Xi = \xi(k_C)$ is a block-diagonal matrix. All blocks of $\Xi$ are either $1 \times 1$ or $2 \times 2$, and are Hermitian. The $1 \times 1$ blocks are at most two for 1D convolution, and at most four for 2D convolution.*

*Proof.* We will treat the case of 1D convolution, and note where the proof changes in a non-trivial manner for 2D. We write $x = \sum_{n=0}^{N-1} a_k c_k$, as in Lemma 4.2. From there we have $Cx = C \sum_{n=1}^{N} a_k c_k = \sum_{n=1}^{N} \lambda_k^{-\mu} a_k c_k$. Then:

$$\|Cx\|^2 = (Cx)^H Cx = c^H Q_N^{-\mu} C^H C Q_N^{\mu} c = \sum_{m=0}^{N-1} \sum_{n=0}^{N-1} \overline{c_m} a_m^H \overline{\lambda_m^{-\mu}} \lambda_n^{-\mu} a_n c_n. \tag{28}$$

The caveat is that while we know that $a_m, a_n$ are unit-length pairwise orthogonal, which would have most terms in the above sum to vanish, it appears we can't commute the $\overline{\lambda_m^{-\mu}} \lambda_n^{-\mu}$ quaternion scalar terms out of the way. We can however proceeding by writing $\lambda_n^{-\mu}$ as a sum of a simplex and a perplex part, with respect to axis $-\mu$ (Ell & Sangwine, 2007), i.e. $\lambda_n = \lambda_n^{\|} + \lambda_n^{\perp}$ (we have dropped the axis superscript in favor of a more clear notation). Eq. 28 follows up as:

$$\sum_{n=0}^{N-1} \overline{c_n} a_n^H |\lambda^n|^2 a_n c_n + \sum_{n=0}^{N-1} \sum_{m \neq n} \overline{c_m} a_m^H (\overline{\lambda_m}^{\|} + \overline{\lambda_m}^{\perp})(\lambda_n^{\|} + \lambda_n^{\perp}) a_n c_n$$

$$= \sum_{n=0}^{N-1} \overline{c_n} |\lambda^n|^2 \|a_n\|^2 c_n + \sum_{n=0}^{N-1} \sum_{m \neq n} \overline{c_m} a_m^H (\overline{\lambda_m}^{\|} + \overline{\lambda_m}^{\perp})(\lambda_n^{\|} + \lambda_n^{\perp}) a_n c_n$$

$$= \sum_{n=0}^{N-1} \overline{c_n} |\lambda^n|^2 c_n + \sum_{n=0}^{N-1} \sum_{m \neq n} \overline{c_m} a_m^H (\overline{\lambda_m}^{\|} \lambda_n^{\|} + \overline{\lambda_m}^{\perp} \lambda_n^{\perp}) a_n c_n + \sum_{n=0}^{N-1} \sum_{m \neq n} \overline{c_m} a_m^H (\overline{\lambda_m}^{\|} \lambda_n^{\perp} + \overline{\lambda_m}^{\perp} \lambda_n^{\|}) a_n c_n =$$

$$= \sum_{n=0}^{N-1} |\lambda^n|^2 |c_n|^2 + \sum_{n=0}^{N-1} \sum_{m \neq n} \overline{c_m} (\overline{\lambda_m}^{\|} a_m^H a_n \lambda_n^{\|} + \overline{\lambda_m}^{\perp} a_m^{\top} \overline{a_n} \lambda_n^{\perp}) c_n + \sum_{n=0}^{N-1} \sum_{m \neq n} \overline{c_m} a_m^H (\overline{\lambda_m}^{\|} \lambda_n^{\perp} + \overline{\lambda_m}^{\perp} \lambda_n^{\|}) a_n c_n =$$

$$= \sum_{n=0}^{N-1} |\lambda^n|^2 |c_n|^2 + \sum_{n=0}^{N-1} \sum_{m \neq n} \overline{c_m} (\overline{\lambda_m}^{\|} a_m^H a_n \lambda_n^{\|} + \overline{\lambda_m}^{\perp} [a_n^H a_m]^H \lambda_n^{\perp}) c_n +$$

$$+ \sum_{n=0}^{N-1} \sum_{m \neq n} \overline{c_m} (\overline{\lambda_m}^{\|} [a_n^{\top} a_m]^H \lambda_n^{\perp} + \overline{\lambda_m}^{\perp} a_m^{\top} a_n \lambda_n^{\|}) c_n. \tag{29}$$

In the above form, the simplex parts commute with eigenvectors $a_n, a_m$, while the perplex parts conjugate-commute with the same eigenvectors. The reason is that all terms of an eigenvector (column of a QFT matrix) share the same axis $\mu$. For $n \neq m$, we have $a_m \perp a_n$ as any QFT matrix is unitary (Prop. 3.3), and $a_m, a_n$ are different columns of a QFT matrix by assumption. Hence, the second sum vanishes.

The third sum in eq. 29 is of special interest, because for each $n \in [0, N-1]$ there will be *at most* one index $m = n'$ over which the term will not vanish. That is because $a_n^{\top} a_m$ is the inner product between $\overline{a_n}$ and $a_m$ (i.e. where the first term is conjugated), so in effect it is a product between a QFT column and another column of the *inverse* QFT (cf. Proposition 3.3). If and only if indices are $n$ and $m = [N - n]_N$, we have $a_n^{\top} a_m = 1$. In other cases, the term vanishes. Consequently, we have:

$$\|Cx\|^2 = \sum_{n=0}^{N-1} |\lambda_n|^2 |c_n|^2 + \overline{c_m} \{ \overline{\lambda_m}^{\|} \lambda_n^{\perp} + \overline{\lambda_m}^{\perp} \lambda_n^{\|} \} c_n, \tag{30}$$

where we use $m = [N - n]_N$. Also, recall that all left eigenvalues $\lambda_n$ are w.r.t. axis $-\mu$.

Note that, if and only if $n = 0$ or $n = N/2$ the second sum will also vanish; these are the columns of the QFT matrix that contain only real-valued terms. For a doubly-block circulant matrix these will be at most *four* by construction. To see this, we can use Proposition 3.3.7, which in effect says that the inner product of the $m^{th}$ and $n^{th}$ column of $Q$ will equal to 1 only for $m = n = 0$ or $N/2$. If the number of elements $N$ is an odd number, only the case $m = n = 0$ is possible. In either case, this coincides with the number of non-zero elements in the diagonal of the permutation matrix $\check{P} = QQ$.

For 2D convolution and a doubly-block circulant matrix of size $N^2 \times N^2$ [3], we obtain an analogous result. We are then interested in the product $(Q_N \otimes Q_N)(Q_N \otimes Q_N)$. This equals to $Q_N Q_N \otimes Q_N Q_N = \check{P} \otimes \check{P}$. The result is another permutation matrix, which will have at most 4 non-zero elements in the diagonal, the number of which will depend to whether $N$ is odd or even. For the second term in eq. 30 we need $m = \varpi([N - i_n, N - j_n]_{N,N})$ with $(i_n, j_n) = \varpi^{-1}(n)$ instead of $m = [N - n]_N$ of the 1D case, so again we have at most one non-vanishing second term for each $n \in [0, N - 1]$.

We can proceed from eq. 30 by rewriting it in the form $\|Cx\|^2 = c^H \Xi' c$, where $\Xi' \in \mathbb{H}^{N \times N}$ is Hermitian. For example, for $N = 6$ we have:

$$\Xi' = \begin{bmatrix} |\lambda_0|^2 & 0 & 0 & 0 & 0 & 0 \\ 0 & |\lambda_1|^2 & 0 & 0 & 0 & \overline{\lambda_1}^{\|} \lambda_5^{\perp} + \overline{\lambda_1}^{\perp} \lambda_5^{\|} \\ 0 & 0 & |\lambda_2|^2 & 0 & \overline{\lambda_2}^{\|} \lambda_4^{\perp} + \overline{\lambda_2}^{\perp} \lambda_4^{\|} & 0 \\ 0 & 0 & 0 & |\lambda_3|^2 & 0 & 0 \\ 0 & 0 & \overline{\lambda_4}^{\|} \lambda_2^{\perp} + \overline{\lambda_4}^{\perp} \lambda_2^{\|} & 0 & |\lambda_4|^2 & 0 \\ 0 & \overline{\lambda_5}^{\|} \lambda_1^{\perp} + \overline{\lambda_5}^{\perp} \lambda_1^{\|} & 0 & 0 & 0 & |\lambda_5|^2 \end{bmatrix},$$

which, by rearranging indices of coefficients and corresponding rows and columns, we have $\Xi'$ which is also Hermitian. For the previous example for $\Xi'$, we would have:

$$\Xi = \begin{bmatrix} |\lambda_0|^2 & 0 & 0 & 0 & 0 & 0 \\ 0 & |\lambda_3|^2 & 0 & 0 & 0 & 0 \\ 0 & 0 & |\lambda_1|^2 & \overline{\lambda_1}^{\|} \lambda_5^{\perp} + \overline{\lambda_1}^{\perp} \lambda_5^{\|} & 0 & 0 \\ 0 & 0 & \overline{\lambda_5}^{\|} \lambda_1^{\perp} + \overline{\lambda_5}^{\perp} \lambda_1^{\|} & |\lambda_5|^2 & 0 & 0 \\ 0 & 0 & 0 & 0 & |\lambda_4|^2 & \overline{\lambda_4}^{\|} \lambda_2^{\perp} + \overline{\lambda_4}^{\perp} \lambda_2^{\|} \\ 0 & 0 & 0 & 0 & \overline{\lambda_2}^{\|} \lambda_4^{\perp} + \overline{\lambda_2}^{\perp} \lambda_4^{\|} & |\lambda_2|^2 \end{bmatrix}.$$

Furthermore, $\Xi$ is also block-diagonal, and all of its blocks are of size $2 \times 2$, with the exception of at most two (four) $1 \times 1$ blocks for 1D (2D) convolution. We can then write the required rearranging of coefficients as a multiplication by a permutation matrix $P$ (not unique, and in general $\neq \check{P}, \tilde{P}$). Hence, $\|Cx\|^2 = c'^H P^\top \Xi' P c'$ $= c^H \Xi c$, where we have switched places for $c, c'$ for clarity of notation. $\square$

Matrix $\Xi$ is important, because it contains sufficient information to compute the spectrum of the convolution. The notation $\xi(k_C)$ stresses that we only need the kernel of the convolution to construct $\Xi$. The structure of the $2 \times 2$ blocks of $\Xi$ will be of the form:

$$\begin{bmatrix} |\lambda_m|^2 & \bar{\lambda}_m^{\|} \lambda_n^{\perp} + \bar{\lambda}_m^{\perp} \lambda_n^{\|} \\ \bar{\lambda}_n^{\|} \lambda_m^{\perp} + \bar{\lambda}_n^{\perp} \lambda_m^{\|} & |\lambda_n|^2 \end{bmatrix}, \tag{31}$$

where $\|$ and $\perp$ refer to simplex and perplex components w.r.t. axis $\mu$ (cf. Appendix, or e.g. Ell & Sangwine (2007)), and $\lambda$ terms are *left* eigenvalues of the convolution *kernel $k_C$*. Its $1 \times 1$ blocks are simply equal to a square-magnitude $|\lambda_m|^2$. Indices $m, n$ must form a pair as described in the Appendix.

---

[3] This is trivially extensible to non-square-shaped filters, i.e. $MN \times MN$.

**Proposition 4.4.** *All singular values can be computed exactly as either the magnitude of a left eigenvalue of $C$, or the square-root of one of the two eigenvalues of the $2 \times 2$ blocks of $\Xi$, again each dependent on a pair of left eigenvalues of $C$. The maximum value of $\|Cx\|^2$ under the constraint $\|x\| = 1$ is given by the square of the largest* right *eigenvalue of $\Xi = \xi(k_C)$.*

*Proof.* The required maximum is equal to the maximum of the Rayleigh quotient $\frac{x^H C^H C x}{x^H x}$, which obtains its maximum for the maximum right eigenvalue (Macías-Virgós et al., 2022, Proposition 3.3-3.4). Now, since $c, x$ are related through a bijection, maximizing the Rayleigh quotient of $C^H C$ is equivalent to maximizing the Rayleigh quotient of $\Xi$.

As $\Xi$ is Hermitian, we know that all of its right eigenvalues are real-valued (and are also left eigenvalues). Due to it being block-diagonal, its right eigenvalues will be the union of the eigenvalues of its blocks, due to Lemma 4.3. As all blocks are either $1 \times 1$ or $2 \times 2$ by Proposition 4.1, the sought right eigenvalue will either be: a) equal to the value of the $1 \times 1$ block, so equal to $|\lambda_n|^2$, where $\lambda_n$ is the left eigenvalue of $C$ that corresponds to the $n^{th}$ column of QFT matrix with axis $\mu$, or b) equal to the right eigenvalues of the $2 \times 2$ Hermitian matrix:

$$\begin{bmatrix} |\lambda_m|^2 & \overline{\lambda_m}^{\|} \lambda_n^{\perp} + \overline{\lambda_m}^{\perp} \lambda_n^{\|} \\ \overline{\lambda_n}^{\|} \lambda_m^{\perp} + \overline{\lambda_n}^{\perp} \lambda_m^{\|} & |\lambda_n|^2 \end{bmatrix}. \tag{32}$$

The right eigenvalues for the latter case are real-valued and can in principle be computed through a complex adjoint mapping and the spectral theorem (Zhang, 1997). (Recall that by $|\lambda_n|$ we denote the left eigenvalue of $C$, produced by taking the $n^{th}$ element of the right QFT with respect to axis *minus* $\mu$ (cf. Proposition 3.3, Lemma 4.3). $\qquad\square$

The above results tell us that we can compute the spectrum of our convolution without building the circulant matrix explicitly. Note that in the non-quaternionic case, $\Xi$ would be diagonal, as off-diagonal terms would vanish, and all $2 \times 2$ blocks would become diagonal. In $\mathbb{H}$, we require the extra step of computing singular values for the $2 \times 2$ blocks.

*Bounding the Lipschitz constant.* In order to set a bound on the spectral norm, we need to be able to move in the opposite direction – from a clipped $\check{\Xi}$ matrix, to new left eigenvalues, back towards the new kernel. The following corollary of the previous propositions is a necessary step (please refer to the appendix for its proof):

**Corollary 4.4.1.** *For any matrix $\Phi$ such that $\Phi^H \Phi = \xi(k_C)$, matrices $C$ and $\Phi$ have the same singular values.*

The choice of $\Phi$ is not unique, and ideally we want a matrix that will be convenient in the context of clipping and reconstruction. We choose $\Phi$ so that it is block-diagonal, with block structure identical to that of $\Xi$. We pick the $1 \times 1$ blocks to be equal to $[\lambda_m]$, and $2 \times 2$ blocks equal to $\begin{bmatrix} \lambda_m^{\|} & \lambda_n^{\perp} \\ \lambda_m^{\perp} & \lambda_n^{\|} \end{bmatrix}$[4]. It is straightforward to confirm that this choice fulfills the requirement set in Corollary 4.4.1. Importantly, this choice of $\Phi$ is related through a bijection with the left eigenvalues of $k_B$. Another bijection exists between left eigenvalues of $k_B$ (cf. corollary 3.6.1) for a given axis, so exact reconstruction of the required clipped convolution is possible.

By making use of the above results, the spectral norm of a given convolutional layer can be clipped as follows: 1) Compute left eigenvalues of the filter using the right QFT, given a choice of $\mu$ (following the results of proposition 3.3). 2) Compute block-diagonal elements of $\Phi$ (cf. corollary 4.4.1). For $1 \times 1$ blocks we simply copy the appropriate left eigenvalue. For $2 \times 2$ blocks we decompose the two left eigenvalues into simplex and perplex parts, as in eq. 31. 3) Compute QSVD for $\Phi$; this can be implemented as QSVDs for the $2 \times 2$ blocks of $\Phi$. Clip singular values and reconstruct $\check{\Phi}$. 4) Reconstruct filters using inverse right QFTs (cf. proposition 3.6). 5) Clip *spatial* range of resulting filters to original range (e.g. (Sedghi et al., 2018, Section 3)). Please see also Fig. 5 for a visual summary of the key steps of the process. All convolutions are clipped and reconstructed with this process. The rest of the linear components are clipped and reconstructed as described in the beginning of this Section.

---

[4]Another choice can be: $\Phi = \Xi^{1/2}$. However it will allow us to reconstruct left eigenvalues only w.r.t their magnitude.

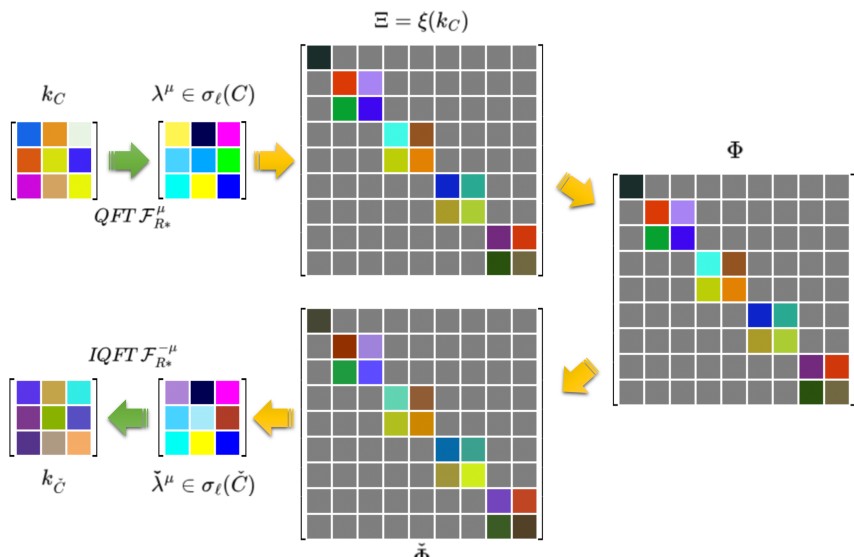

Figure 5: A visual summary of the procedure that forms the backbone of the Lipschitz constant-bounding method, enabled with our results. Given a quaternionic convolution kernel $k_C$ (top left), we compute left eigenvalues $\lambda^\mu$ via the QFT with axis $\mu \in \mathbb{H}$. From our left eigenvalues we construct matrix $\Xi$, then matrix $\Phi$ (see in-text for details). Crucially, matrix $\Phi$ has the same singular values with doubly-block convolutional matrix $C$, but is much easier to construct and manipulate due to its block diagonal structure, with blocks sized either $1 \times 1$ or $2 \times 2$. Spectrum manipulation is thus performed over $\Phi$ to obtain $\check{\Phi}$, and in turn we obtain new left eigenvalues $\check{\lambda}^\mu$. Finally, we apply IQFT to obtain the quaternionic convolution kernel that corresponds to the manipulated/clipped convolution $k_{\check{C}}$. The result is a method that is more economical in terms of space and time requirements by orders of magnitude, related a process that would be oblivious to these results.

*Results and Discussion: Complexity of Space and Time Requirements.* Direct clipping using QSVD is very expensive both in terms of space and time complexity. The input to QSVD is the doubly-block circulant matrix that corresponds to the convolution filter, and it consists of $N^4$ quaternionic elements; our approach does not require computing the doubly-block circulant matrix at all. In terms of time complexity, direct clipping using QSVD requires $\mathcal{O}(N^6)$, the cost of SVD over $C$. Regarding our method, for the forward and inverse QFT, we need 4 FFTs, as each QFT is computed via 2 standard FFTs (Ell & Sangwine, 2007). Also, we require $\approx N^2/2$ eigenvalue decompositions for the $2 \times 2$ blocks of $\Phi$ (depending on how many $1 \times 1$ blocks exist, which in turn depends on whether $N$ is odd or even). Hence, total complexity is in the order of $\mathcal{O}(N^2 log N)$.

*Computing Singular Values Clipping Time Comparison versus Brute-force QSVD.* We have tested runtime of our method versus the brute-force QSVD approach, over a set of random filters. Results can be examined in Table 1, where we compare runtimes using the method we described, versus the brute-force method (direct clipping). Computational times to either only compute singular values, or perform the full clipping and reconstruction process, are also reported. Concerning the size parameter in a practical application, note that while small convolutional kernel sizes as much as $3 \times 3$ or $5 \times 5$ are prevalent in both real-valued and hypercomplex network architectures, when we express convolution in terms of a circulant or doubly-block circulant matrix we need to take into account the *zero-padded* version of the kernel. In simple terms, when $Cx$ expresses a convolution over input $x \in \mathbb{H}^N$, circulant $C$ must match the dimension of the input, so $C \in \mathbb{H}^{N \times N}$. Hence, when comparing resource requirements for different sizes of convolution, larger magnitudes for $N$ are more relevant to real-world practice. On operation sizes $N \geq 128$, using the brute-force method is practically impossible due to both space and time constraints. On the contrary, our method scales well on larger sizes. A visualization of a clipped and reconstructed filter can be examined in the Appendix.

Table 1: Runtime comparison for computation of Quaternionic Convolution Singular Values and Spectral Norm Clipping & Reconstruction. Convolution kernel size refers to $2D$ kernel side length. Average CPU time in milliseconds is reported; an exception are the fields containing an asterisk ($\star$), which denote that required runtime is impractical (measured in hours). Our results enable fast and exact carrying out of both tasks.

| | | Required time (in ms) ↓ | | | | | |
|---|---|---|---|---|---|---|---|
| Convolution kernel size | | 4 | 8 | 16 | 32 | 64 | 128 |
| Singular Value Computation | ours | 4 | 16 | 80 | 210 | 708 | $1,795$ |
| | brute-force | 5 | 95 | $1,244$ | $69,319$ | $\star$ | $\star$ |
| Clipping & Reconstruction | ours | 5 | 20 | 113 | 280 | $1,119$ | $2,590$ |
| | brute-force | 8 | 121 | $1,866$ | $150,000$ | $\star$ | $\star$ |

*Numerical stability test.* We have tested the spectral norm clipping method in terms of numerical stability. To this end, given an "original" quaternionic convolution kernel $A \in \mathbb{H}^{N \times N}$, we apply over it multiple iterations of clipping & reconstruction. (That is, in the sense of starting from a specific kernel, we inject noise, clip & reconstruct, then re-inject noise, clip & reconstruct, and so on). We clip to a bound equal to a given percentage of the original maximum singular value. The original kernel is constructed as a sample from a standard Normal. We have run the experiment over a set of different values for the variance of the injected noise ($\sigma^2 = 1, 3, 5$), over different clipping percentages ($80\%, 90\%, 99\%$), and over different quaternionic convolution sizes ($3, 5, 7, 9$).

We have run the experiment over a total of 100 iterations of clipping & reconstruction, using either the result-enabled method, or the "brute-force" method. In all cases, the numerical difference measured as sum of absolute values between the results of the two methods was steadily under $10^{-11}$ (we use the same random seed, and the only difference is between the clipping & reconstruction method applied).

*Application on a Neural Network.* We test a ResNet32 architecture on CIFAR10. We replaced standard convolutional layers with depthwise convolutions and pointwise convolutions (Chollet, 2017) on the quaternionic domain. We report the results of training both architecture variants considered (QResNet32-small and QResNet32-large), with and without clipping, in Table 2. We observe that *clipping always boosts final performance, regardless of the architecture considered.*

Table 2: Comparison of final test accuracy, when setting a bound on the Lipschitz constant (w/ clipping) versus not (w/o clipping). Figures for two considered architectures are reported, as statistics over 5 runs. Spectral norms are clipped to value $c = 1.0$.

| architecture | w/o clipping | w/ clipping |
|---|---|---|
| QResNet32-small | $81.43\% \pm 0.75$ | $85.80\% \pm 0.38$ |
| QResNet32-large | $81.78\% \pm 1.04$ | $86.32\% \pm 0.33$ |

To further understand the dynamics of the clipping procedure, we present the loss and accuracy curves in Figure 6 for the case of *QResNet32-small*, where the per epoch loss and accuracy are reported for both the with and without clipping variants, showing mean and standard deviation values over 5 runs. As we can see, the clipping impact is mostly realized when the scheduling step occurs (i.e., at 40 epochs), indicating that it enables a finer tuning of the network.

*Lipschitz constant bounding tests on CIFAR10.* We follow (Sedghi et al., 2018) and test a ResNet32 architecture. In our model, 3 convolutional blocks, each one consisted of 5 basic residual blocks, are considered, followed by a fully connected layer. Between each block a downsampling operation is used (with a strided convolution). An initial convolutional layer transforms the $3d$ image input into a $16d$ feature tensor.

For our case, we replaced the convolutional layers of basic residual with depthwise convolutions (Chollet, 2017; DeMagistris et al., 2022). Between the blocks, pointwise convolutions are also added to change the

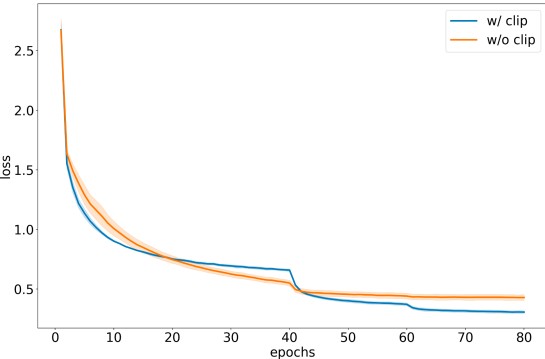 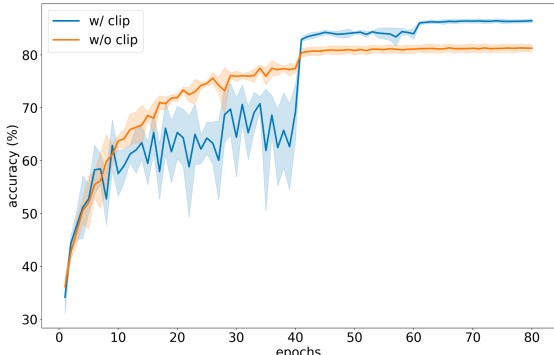

Figure 6: Per-epoch analysis of the impact of using the proposed technique. Training loss (left) and test accuracy (right) curves are reported. Clipping using the method enabled by our results always leads to a clear performance boost.

number of dimensions and thus each block can safely proceed with depthwise only operations, using the same number of channels. Of course, these depthwise and pointwise operations are replaced with the quaternionic alternatives. Only the first $3 \times 3$ convolution and the last fully connected layer are retained as they are. This model is referred to as *QResNet32-small* and has only $98k$ parameters. An alternative has also been explored, where between each depthwise convolution of the basic residual block, an extra pointwise convolutional layer is intervened. Its functionality is to assist in the mixing of the channels, at the cost of greatly increasing the number of parameters. Specifically, this version, dubbed as *QResNet32-large* has approximately $454k$ parameters.

Training was performed on CIFAR10, where each model was trained for 80 epochs. Initial learning rate was 0.01 and was subsequently dropped by /10 at 40 and 60 epochs. Clipping is performed every 100 iterations. A summary of architecture and optimization details can be found in the Appendix (Section F).

We show comparisons in Table 2 and Figure 6 in Section 4 of the main text. Also, to understand how clipping may affect the result, we report the singular values of each quaternionic layer in Figure 7, for training the *QResNet32-small* architecture without clipping. For visualization purposes, each layer is depicted by a mean value and the standard deviation of the corresponding singular values of the layer.

We can observe that a) clipping always boosts final performance, regardless of the architecture considered; b) the overall performance of the two architectures is very close. Hence, we continue our experimentation with one of the variants.

Next, we investigated the impact of the clipping value $c$. The results for a QResNet32-small architecture are summarized in Table 3, over 5 runs for each setting. As we can see, the choice of the clipping value has an impact on overall performance. Specifically, choosing a value from 0.25 to 0.5 seems to be more beneficial, while lower or higher values over-"prune" or are too modest to provide notable results.

Table 3: Impact of clipping value $c$ on overall model performance.

| $c = 0.10$ | $c = 0.25$ | $c = 0.50$ | $c = 1.00$ | $c = 2.00$ |
|---|---|---|---|---|
| $72.74\% \pm 3.34$ | $82.83\% \pm 1.02$ | $85.88\% \pm 0.52$ | $85.80\% \pm 0.38$ | $84.21\% \pm 0.28$ |

Another aspect that we have tested is the impact of the choice of the axis $\mu$ of the QFT. This acts as a hyperparameter, as we need QFT to compute left eigenvalues, and reconstruct the filter from the new left eigenvalues after singular value clipping. In theory, its impact on performance should be minimal. Towards this, we randomly generated a total of 20 different runs, with a different value for $\mu$ on each. Test accuracy curves for are visualized in Figure 8. As we can see, clipping is equally effective, regardless of our choice of the $\mu$ axis.

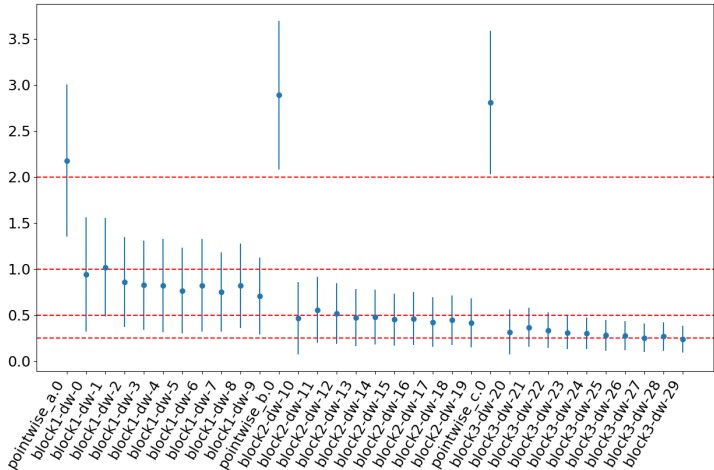

Figure 7: Distribution of the singular values for each layers, when training a *QResNet32-small* architecture. The reported values are for the final model, after 80 epochs.

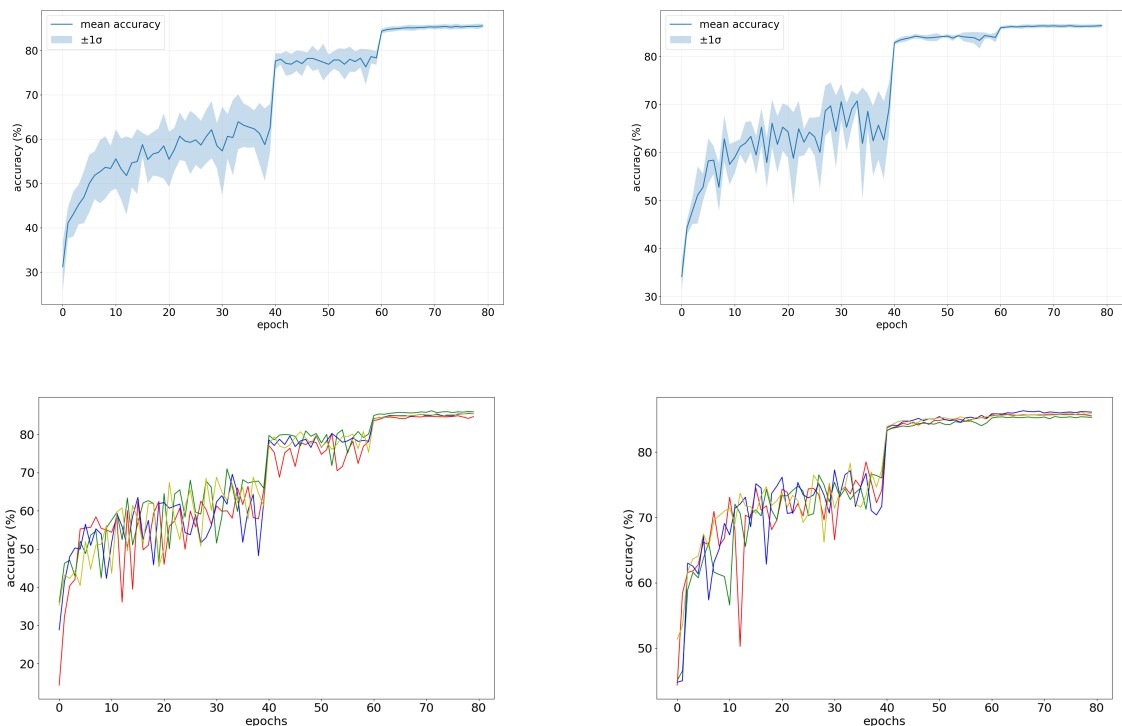

Figure 8: Test accuracy curves during the training for the *QResNet32-small* architecture. A set of 20 random initializations are considered, using different values for axis $\mu$. A random selection of runs and mean / standard deviation values for all runs are visualized on the top and bottom rows respectively. Experiments for two different clipping values, $c = 0.5$ and $c = 1.0$ are shown, respectively corresponding to the left and right column. Regardless of model and clipping parameters, clipping is equally effective in all cases.

Note also that the non-quaternionic equivalent, with only depthwise convolutions in the residual blocks, is under-performing, achieving 75.41% test accuracy, even when using more parameters, namely $151K$ parameters in total. We explain this interesting result as follows: Cascades of depthwise convolutions in $\mathbb{R}$ are much less expressive than their counterpart in $\mathbb{H}$, because the former represent series of convolutions over a single real-valued channel. The latter however, do comprise a form of "channel mixing" under the hood, due to the definition of the Hamilton product. Hence, the quaternionic depthwise layers stand as an interesting trade-off between the completely independent convolutions of real-valued depthwise convolution, and full convolution, taking into account correlation between all possible pairs of channels.

*Lipschitz constant bounding tests on an additional backbone and additional benchmarks.* In order to further test the validity of our results, we have run additional tests using an adaptation of the VGG16 backbone, following a similar process for its quaternionization as for the aforementioned ResNet variants. In particular, we have run tests on the additional tests Fashion MNIST and CIFAR100, using the large ResNet variant. Results for these sets, as well as results for tests using VGG16 can be examined in Table 4. In all of these cases we have run versions using clipping vs without clipping, as done with our CIFAR10 tests. Note also the low variance of reported accuracy w.r.t. different runs, which highlights the stability of the results-enabled clipping method.

Table 4: Additional results comparing final test accuracy, when setting a bound on the Lipschitz constant (w/ clipping) versus not (w/o clipping). Figures for tests on datasets CIFAR100 and Fashion MNIST are reported, as well as over an additional quaternionized VGG16 backbone (QVGG16). Statistics are reported over 5 runs. Spectral norms are clipped to value $c = 1.0$.

| dataset / architecture | w/o clipping | w/ clipping |
|---|---|---|
| Fashion MNIST / QResNet32-large | $89.73\% \pm 0.86$ | $92.36\% \pm 0.27$ |
| CIFAR100 / QResNet32-large | $56.05\% \pm 1.32$ | $61.15\% \pm 0.19$ |
| CIFAR10 / QVGG16 | $90.38\% \pm 0.12$ | $90.80\% \pm 0.02$ |

## 5 Conclusion

We have presented a set of results concerning the properties and relation between the Quaternion Fourier Transform and Quaternion Convolution, with respect to their formulations in terms of Quaternionic matrices. This relation has been well-known to be very close and important on the real and complex domains, and already used in a wide plethora of learning models and signal processing methods. With our results, we have shown how and to what extend these properties generalize to the domain of quaternions, opening up the possibility to construct analogous models built directly on quaternionic formulations. Given the ubiquitous character of linear transformations, and consequently, matrix representations in virtually every subfield of machine learning, our results pave the way for new, expressive data science models (Qin et al., 2023; Guo et al., 2025; Parcollet et al., 2020; Yue et al., 2025; Miao et al., 2024). As proof of concept, we have presented a method for bounding of the Lipschitz constant using our theoretical results; the method enables spectral norm bounding in quaternionic NNs, previously impossible in practice. Lipschitz constant bounding and determination is important in a number of important applications, and important in different classes of models, including Wasserstein GANs (Petzka et al., 2018), Plug-n-Play networks (Ryu et al., 2019), or Deep Equilibrium Networks (Pabbaraju et al., 2021); hence, one direct line of research involving our results could envisage generalizations of these applications in quaternionized versions of these models.

Let us refer to two more concrete examples of perspectives or application of our results:

- *Richer observation models.* Assuming an observation model of the form $g = h * f + n = Hf + n$ (see for example Chantas et al. (2009) or Hidalgo-Gavira et al. (2019)), we can now consider casting all components as quaternionic. Assuming a likelihood model $p(g|f) \propto \exp\{-\frac{\beta}{2}||g - Hf||^2\}$ and a prior $p(\Phi f|a) \propto \exp\{-\frac{a}{2}||\Phi f||^2\}$, where $\Phi$ is cast as a convolution, a solution in $\mathbb{R}$ is known to include operations that require results on addition and multiplication of circulant matrices. Hence, with the current work, this direction can be envisaged, making direct use of e.g. Proposition 3.7.

- *Applications to adaptations of a wide range of NNs to $\mathbb{H}$.* Convolution, as a linear operation, can and has been used in tandem with layers of other types. A good example here is hybrid architectures that have emerged relatively recently. Models like the Convolutional Vision Transformer replace the linear projections for queries, keys, and values with convolutional operations. Other approaches, such as the Conformer for speech recognition, place a dedicated convolution module within the Transformer block to capture local correlations alongside the self-attention mechanism's global context (Gulati et al., 2020). The Mamba architecture also includes a convolution layer as part of its internal block to extract local patterns before its Structured-State Space Model focuses on long-range dependencies. Other more direct "hybrids" with traditional convnets also exist, like ConvMamba or VisionMamba (Gu & Dao, 2023; Yu & Wang, 2025). Graph convolution is another widely used example of fusion with convolution (Zhang et al., 2019). With the results we present in Section 4, Lipschitz-constant bounding on quaternionized versions of these architectures is a straightforward step.

Future research will shed more light over these topics.

## Acknowledgments

The authors would like to thank the TMLR action editor and the anonymous reviewers for their constructive feedback and suggestions.

## Dedication

This paper is dedicated in memory of the late Dr. N.P. Galatsanos.

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

## Appendix

## A  Reconstruction Visualization of Quaternionic Doubly-block Circulant Matrix

In Section 4 ("Proof-of-concept Application") of the main text, we refer to a kernel for our experiments which was produced using the code that follows. We report the code in the interest of reproducibility of the result. We used the "Quaternion" Python package (Boyle, 2018) (as well as for all implementations required for this paper):

```python
import numpy as np
import quaternion
# Using np.quaternion from
# https://github.com/moble/quaternion
C_filter = np.zeros([N, N], dtype=np.quaternion)
for i in range(N):
 for j in range(N):
  lm1 = np.cos(i)
  lm2 = np.sin(i)
  lm3 = np.cos(i+.3)
  lm4 = np.sin(i-.2)
  C_filter[i, j] = np.quaternion(
    i+lm1/N, j+lm2/N, i*j*lm3, i*j*lm4
    )
```

In Figure 9 we show an example result of clipping and reconstruction using our method (the proof-of-concept method of Section 4 applied over a single filter) versus applying directly the QSVD ("brute-force") on the corresponding doubly-block circulant matrix. The test filter used is a $32 \times 32$ quaternionic convolution. For reference, the mean (maximum) of the singular values of the original kernel is 3005.2 (238519.9) and the clipping threshold was set to 4000. We have used $\mu = (\boldsymbol{i} + \boldsymbol{j} + \boldsymbol{k})/\sqrt{3}$ as the QFT axis, following (Ell & Sangwine, 2007).

Both singular values and the reconstruction result are exact using our method. The discrepancy of point-to-point difference between our reconstruction and brute-force is in the ballpark of $10^{-11}$ units of magnitude, at worst (cf. last row in Figure 9). In terms of time requirements, our method required 280 milliseconds to obtain an exact reconstruction, compared to $\sim 3$ minutes for the brute-force method.

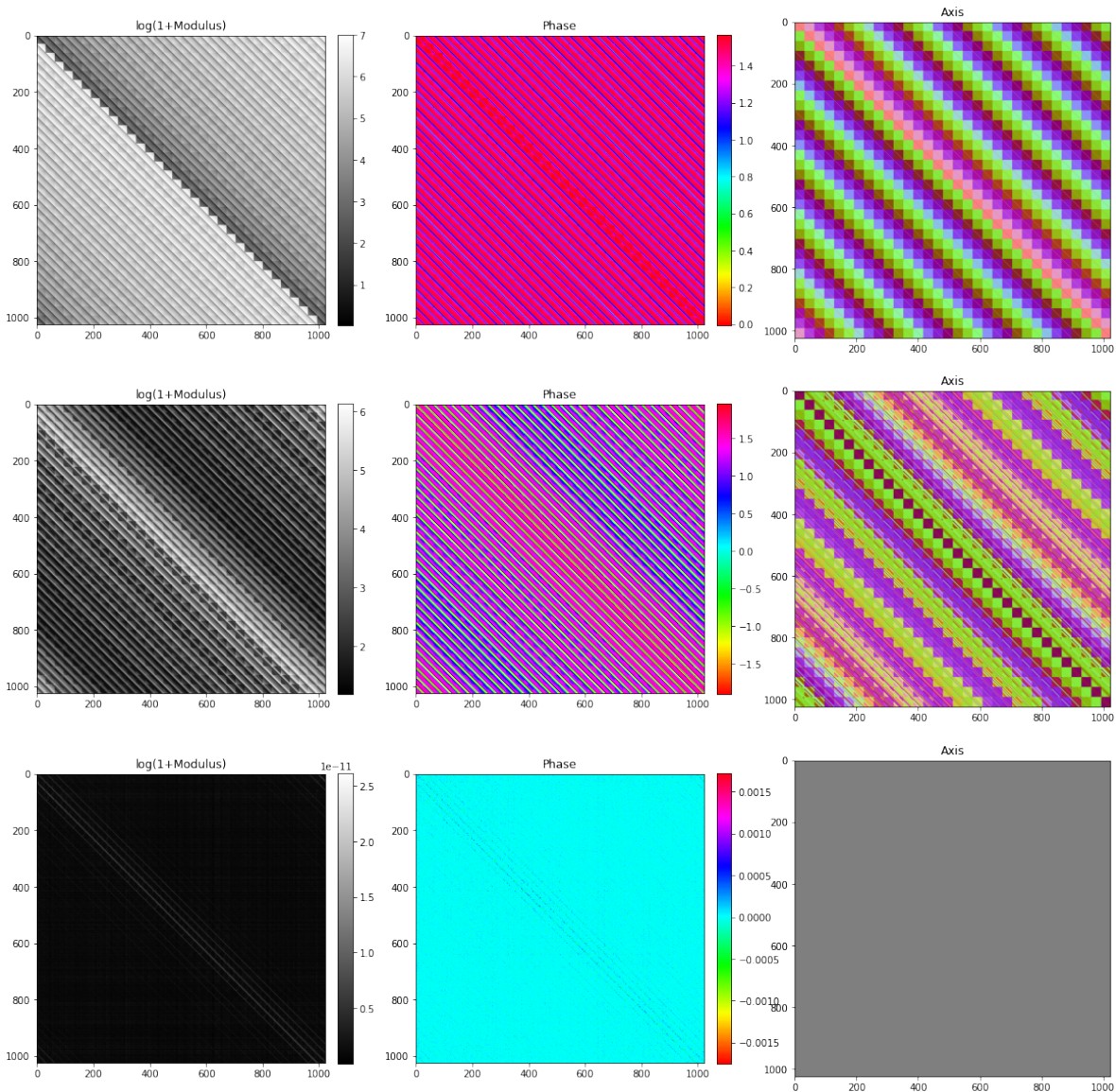

Figure 9: Constraining the spectral norm using QSVD directly on the doubly-block circulant matrix (referred to as "brute-force" in text) vs our method. From top to bottom, we see: the original kernel, the clipped kernel using our method, the point-to-point difference between our result and the brute-force method. From left to right, we see log-magnitudes, phases and axes of respective doubly-block circulant matrices (cf. (Ell & Sangwine, 2007)). The brute-force method is considerably more expensive in terms of space as well as time requirements, both by orders of magnitude. For this $32 \times 32$ kernel, our method required 280 milliseconds to obtain an exact reconstruction, while brute-force took over 2 minutes. (see text of Section 4 in the main text for more details).

## B    Notation conventions

Throughout the text, we have used the following conventions. Capital letters $M, N$ will refer to matrix dimensions, input or filter size. On the "Proof-of-concept" Section 4, $N$ refers to the side size of the input and output images, or equivalently, the zero-padded kernels. Letters $\mu, \nu$ refer to pure unit quaternions, usually used as axes of a QFT. A QFT matrix is written as $Q_N^\mu$, referring to the matrix with side $N \times N$ and reference to axis $\mu$. Letters $\kappa, \lambda$ refer to left eigenvalues. They will be used with an exponent, $\lambda^\mu$, which

is to refer to a left eigenvalue with respect to axis $\mu$. When we want to state that a left eigenvalue is related to a specific column of a QFT matrix as its eigenvector, we add a subscript, $\lambda_i^\mu$. Our "by default" ordering is hence with respect to the order of QFT columns (and not e.g. left eigenvalue magnitude). A vector or set of left eigenvalues is denoted boldface, $\boldsymbol{\lambda}^\mu$. A conjugate element is denoted by a bar ($\bar{p}$). An element that has been produced after singular value clipping in the sense discussed in Section 4 is denoted with a "check" character on top: $\check{\Sigma}, \check{\Xi}$.

Indexing, if not stated otherwise, is by default considered to be zero-indexed and "modulo-$N$" (so index 0 is the same as index $N$, or $-1$ is the same as $N-1$ and so on). This convention is used in order to match the $w_N^\mu$ exponents definition of the QFT (eq. 5), which is very important for most of the subsequent theoretical results.

## C  Additional Results and Notions from Quaternion Algebra

*Cayley-Dickson form and symplectic decomposition*: A quaternion may be represented as a complex number with complex real and imaginary parts, in a unique manner. We write

$$q = A + B\boldsymbol{j}, \tag{33}$$

with

$$A = q_1 + q_2\boldsymbol{i}, B = q_3 + q_4\boldsymbol{i}.$$

An analogous operation can be performed for quaternion matrices, which can be written as a couple of complex matrices (Zhang, 1997). This scheme can be easily generalized to using any other couple of perpendicular imaginary units $\mu_1, \mu_2$ instead of $\boldsymbol{i}, \boldsymbol{j}$. In that, more general case, it is referred to as a symplectic decomposition (Ell & Sangwine, 2007), which is essentially a change of basis from $(1, \boldsymbol{i}, \boldsymbol{j}, \boldsymbol{k})$ to new imaginary units $(1, \mu_1, \mu_2, \mu_3)$. We can in general write

$$p = p^\| + p^\perp,$$

where $p^\| = p_0 + p_1\mu_1$ and $p^\perp = (p_2 + p_3\mu_1)\mu_2 = p_2\mu_2 + p_3\mu_3$. Quaternions $p^\|$ and $p^\perp$ are referred to as the parallel / simplex part and the perpendicular / perplex part with respect to some basis $(1, \mu_1, \mu_2, \mu_3)$. Given symplectic decompositions for quaternions $p, q$, their parts may commute or *conjugate-commute* (Ell & Sangwine, 2007, Section 7C). Note the conjugate operator over $p^\|$ in the second relation:

$$p^\| q^\| = q^\| p^\|, \qquad p^\| q^\perp = q^\perp \bar{p}^\|.$$

*Quaternion Singular Value Decomposition*: (Zhang, 1997) and (Le Bihan & Mars, 2004) have been among the first works to discuss the extension of the SVD into the quaternionic domain, dubbed Singular Value Decomposition for Quaternion matrices or simply Quaternion Singular Value Decomposition (QSVD). (Sangwine & Le Bihan, 2006) have proposed a bidiagonalization-based implementation of QSVD.

The fundamental proposition is that, for any matrix $A \in \mathbb{H}^{M \times N}$, or rank $r$, there exist two quaternion unitary matrices $U$ and $V$ such that

$$A = U\Sigma V^H = U \begin{bmatrix} \Sigma_r & 0 \\ 0 & 0 \end{bmatrix} V^H, \tag{34}$$

where $\Sigma_r$ is a real diagonal matrix and has $r$ strictly positive entries on its diagonal (Le Bihan & Mars, 2004, Proposition 10). The elements of the diagonal are termed singular values. Matrices $U \in \mathbb{H}^{M \times M}$ and $V \in \mathbb{H}^{N \times N}$ are made up of columns that are each left and right eigenvectors of $S$. Note that, perhaps confusingly, the adjectives "left" and "right" are not corresponding to the "left" and "right" spectra of the matrix. *Both* sets of eigenvectors correspond to singular values which are in turn rather more closely related to the right spectrum. We can see this by considering $A^H A$ or $A A^H$, which starting from eq. 34 equal to $V\Sigma^2 V^H$ and $U\Sigma^2 U^H$ respectively. Consequently, we have $A^H A V = V\Sigma^2$ and $A A^H U = U\Sigma^2$, which solve the *right* eigenvalue problem, $\tilde{A}x = x\lambda$.

Hence, the singular values are the non-zero *right* eigenvalues of either $A^H A$ or $A A^H$. As these must be in $\mathbb{R}$, as eigenvalues of Hermitian matrices, they are also *left* eigenvalues. Left eigenvalues that are not right eigenvalues may very well exist, even for Hermitian matrices (Zhang, 1997, Example 5.3).

# D Quaternionic Convolution and Quaternionic Fourier Transform

## D.1 Quaternionic Convolution

In the continuous case, quaternionic convolution is defined as (Bahri et al., 2013):

$$(f * g)(x) = \int_V f(u)g(x - u)du, \tag{35}$$

and cross-correlation (Ell et al., 2014) as:

$$(f \star g)(x) = \int_V \overline{f(u)}g(x + u)du, \tag{36}$$

where $V = \mathbb{R}, \mathbb{R}^2$ for 1D and 2D convolution/correlation respectively. Similarity of the above to the well-known non-quaternionic formulae is evident. A number of useful properties of non-quaternionic convolution also hold for quaternionic convolution. These properties include linearity, shifting, associativity and distributivity (Bahri et al., 2013). Perhaps unsurprisingly, quaternionic convolution is non-commutative ($f*g \neq g*f$ in general), and conjugation inverses the order of convolution operators ($\overline{f * g} = \overline{g} * \overline{f}$). In practice, discrete versions of 1D convolution and correlation are employed. In this work we will use the following definitions of 1D quaternion convolution (Ell et al., 2014, Section 4.1.3):

$$(h_L * f)[n] = \sum_{n=0}^{N-1} h_L[i]f[n - i], \tag{37}$$

$$(f * h_R)[n] = \sum_{n=0}^{N-1} f[n - i]h_R[i], \tag{38}$$

where the difference between the two formulae is whether the convolution kernel elements multiply the signal from the left or right. Extending to 2D quaternion convolution, two of the formulae correspond precisely to left and right 1D convolution:

$$(h_L * f)[n, m] = \sum_{n=0}^{N-1}\sum_{m=0}^{M-1} h_L[i, j]f[n - i, m - j], \tag{39}$$

$$(f * h_R)[n, m] = \sum_{n=0}^{N-1}\sum_{m=0}^{M-1} f[n - i, m - j]h_R[i, j], \tag{40}$$

while a third option is available, in which convolution kernel elements multiply the signal from both left and right (Ell et al., 2014):

$$(h_L \prec f \succ h_R)[n, m] =$$
$$\sum_{n=0}^{N-1}\sum_{m=0}^{M-1} h_L[i, j]f[n - i, m - j]h_R[i, j]. \tag{41}$$

This last variant, referred to as bi-convolution (Ell et al., 2014), has been used to define extension of standard edge detection filters (Sobel, Kirsch, Prewitt) for color images. A variation of bi-convolution has also been used in (Zhu et al., 2018) as part of a quaternion convolution neural network, locking $h_L = \overline{h}_R$ and adding a scaling factor.

We introduce circular variants to the above formulae. Circular left convolution is written as:

$$(h_L * f)[n] = \sum_{n=0}^{N-1} h_L[i]_N f[n - i], \tag{42}$$

where $[\cdot]_N$ denotes modulo-$N$ indexing (Jain, 1989).

### D.2 Quaternionic Fourier Transform

For $1D$ signals we have the definition[5] of the left- and right-side QFT:

$$F_L^\mu[u] = \frac{1}{\sqrt{N}} \sum_{n=0}^{N-1} e^{-\mu 2\pi N^{-1} nu} f[n], \tag{43}$$

$$F_R^\mu[u] = \frac{1}{\sqrt{N}} \sum_{n=0}^{N-1} f[n] e^{-\mu 2\pi N^{-1} nu}, \tag{44}$$

where $\mu$ is an arbitrary pure unit quaternion that is called the *axis* of the transform. For 2D signals, these formulae are generalized to the following definitions (Ell & Sangwine, 2007):

$$F_L^\mu[u, v] = \frac{1}{\sqrt{MN}} \sum_{m=0}^{M-1} \sum_{n=0}^{N-1} e^{-\mu 2\pi (M^{-1} mv + N^{-1} nu)} f[n, m], \tag{45}$$

$$F_R^\mu[u, v] = \frac{1}{\sqrt{MN}} \sum_{m=0}^{M-1} \sum_{n=0}^{N-1} f[n, m] e^{-\mu 2\pi (M^{-1} mv + N^{-1} nu)}. \tag{46}$$

We shall also employ the short-hand hand notation $\mathcal{F}_X^\mu\{g\}$ with $X = \{L, R\}$ to denote the left or right QFT with axis $\mu$ of a signal $g$. The inverse transforms will be denoted as $\mathcal{F}_X^{-\mu}$, as we can easily confirm that $\mathcal{F}_X^\mu \circ \mathcal{F}_X^{-\mu} = \mathcal{F}_X^{-\mu} \circ \mathcal{F}_X^\mu$ is the identity transform. Asymmetric definitions are also useful, and we will use $\mathcal{F}_{X*}^\mu$ to denote a transform with a coefficient equal to 1 instead of $1/\sqrt{N}$ and $1/\sqrt{MN}$.

### D.3 Convolution theorem in $\mathbb{H}$

In the complex domain, the convolution theorem links together the operations of convolution and the Fourier transform in an elegant manner. For most combinations of adaptation variants of convolution and Fourier transform in the quaternionic domain, a similar theorem is not straightforward, if at all possible (e.g. (Cheng & Kou, 2019)). (Ell & Sangwine, 2007) have proved the following formulae for right-side convolution (we have changed equations to represent n-dimensional signals in general):

$$\mathcal{F}_L^\mu\{f * h\} = \sqrt{N}\{F_{1L}^\mu[u] H_L^\mu[u] + F_{2L}^\mu[u]\mu_2 H_L^{-\mu}[u]\}, \tag{47}$$

$$\mathcal{F}_R^\mu\{f * h\} = \sqrt{N}\{F_R^\mu[u] H_{1R}^\mu[u] + F_R^{-\mu}[u] H_{2R}^\mu[u]\mu_2\}, \tag{48}$$

which by symmetry are complemented by the following formulae for left-side convolution:

$$\mathcal{F}_L^\mu\{h * f\} = \sqrt{N}\{H_{1L}^\mu[u, v] F_L^\mu[u, v] + H_{2L}^\mu[u, v]\mu_2 F_L^{-\mu}[u, v]\}, \tag{49}$$

$$\mathcal{F}_R^\mu\{h * f\} = \sqrt{N}\{H_R^\mu[u] F_{1R}^\mu[u] + H_R^{-\mu}[u] F_{2R}^\mu[u]\mu_2\}. \tag{50}$$

On the above formulae, transforms $F_L^\mu, H_R^\mu$ are decomposed as:

$$F_L^\mu = F_{1L}^\mu + F_{2L}^\mu \mu_2, F_R^\mu = F_{1R}^\mu + F_{2R}^\mu \mu_2,$$

$$H_L^\mu = H_{1L}^\mu + H_{2L}^\mu \mu_2, H_R^\mu = H_{1R}^\mu + H_{2R}^\mu \mu_2,$$

where we use symplectic depositions (cf. section C) with respect to the basis $(1, \mu, \mu_2, \mu\mu_2)$. Note that $\mu_2$ is an arbitrary pure quaternion conforming to $\mu \perp \mu_2$.

---

[5]As different transforms are obtained by choosing a different axis $\mu$, we have chosen to denote the axis explicitly in our notation. This formulation is slightly more generic than the one proposed by Ell & Sangwine (2007), and we have the correspondence of $F^{-X}$ (their notation) to $F_X^{-\mu}$ (our notation).

# E  Additional Proofs

**Lemma 4.1** *For any pure unit $\mu \in \mathbb{H}$ and any $N \in \mathbb{N}$, the right span of the columns of $Q_N^\mu$ is $\mathbb{H}^N$. For any pure unit $\mu \in \mathbb{H}$ and any $M, N \in \mathbb{N}$, the right span of the columns of $Q_M^\mu \otimes Q_N^\mu$ is $\mathbb{H}^{MN}$.*

*Proof.* Let $c \in \mathbb{H}^N$, and let $x = \sum_{n=1}^N a_k c_k$ (Le Bihan & Mars, 2004, definition 2), where $a_k$ is the $k^{th}$ column of $Q_N^\mu$. In matrix-vector notation, we have $x = Q_N^\mu c$. But $Q_N^\mu$ is invertible for any $\mu, N$ (Proposition 3.3), so $c = Q_N^{-\mu} x$. Thus, for any $x \in \mathbb{H}^N$, we can compute right linear combination coefficients $[c_1 \ c_2 \cdots c_N]^T = c$. The proof for $Q_M^\mu \otimes Q_N^\mu$ is analogous.

(An analogous result holds also for the left span; it suffices to take $c^H = Q_N^{-\mu} x^H$). $\qquad\square$

**Lemma 4.2** *Let $A \in \mathbb{H}^N$ be block-diagonal. The left eigenvalues of each block of $A$ are also left eigenvalues of $A$. The right eigenvalues of each block of $A$ are also right eigenvalues of $A$.*

*Proof.* For a block $A_{ab}$ that is situated in the submatrix of $A$ within rows and columns $a : b$, take $[0 \ \cdots \ 0 \ x \ 0 \cdots 0]^T$ as an eigenvector of $A$, where $x$ is an eigenvector of $A_{ab}$. $\qquad\square$

**Corollary 4.4.1** *For any matrix $\Phi$ such that $\Phi^H \Phi = \xi(k_C)$, matrices $C$ and $\Phi$ have the same singular values.*

*Proof.* We can write the change of basis in eq. 28 as $Q_N^{-\mu} C^H C Q_N^\mu$, and after we combine it with a rearraging of rows and columns encoded as a left-right multiplication by permutation matrix $P$, we have

$$\Xi = P^T Q_N^{-\mu} C^H C Q_N^\mu P = (Q_N^\mu P)^{-1} (C^H C) Q_N^\mu P,$$

so $\Xi$ and $C^H C$ are similar matrices, which entails that they have the same right eigenvalues (Zhang, 1997, Section 7). The same holds for $\Phi^H \Phi$ and $C^H C$, by our assumption that $\Phi^H \Phi = \xi(k_C)$. The singular values of $\Phi$ are the (non-zero) right eigenvalues of $\Phi^H \Phi$, and the singular values of $C$ are the (non-zero) right eigenvalues of $C^H C$, which proves the corollary. $\qquad\square$

# F  Model architecture and Optimization details for experiments on CIFAR-10

*Datasets and preprocessing.* All initial experiments are conducted on CIFAR-10. Training images are augmented with random horizontal flips and random $32 \times 32$ crops after 4-pixel zero-padding. Inputs are converted to tensors and normalized channel-wise using the widely-adopted ImageNet statistics. Evaluation uses the non-augmented (only channel-wise normalization) pipeline.

*Model.* We use a CIFAR-style ResNet-32 topology with three stages of residual blocks and increasing channel width. Each stage has 5 residual blocks. Each block applies a $3 \times 3$ convolution, batch normalization, and ReLU; downsampling between stages is performed by strided convolutions. The network ends with global average pooling and a linear classifier. We then "quaternionize" the model by replacing the typical $3 \times 3$ convolution on the residual blocks with a depthwise quaternionic one. The input convolution layer is not replaced by a quaternionic one. To re-enable (non-quaternionic) channel mixing, we have a variant where a pointwise real convolution follows each introduced quaternionic depthwise convolution at the cost of more parameters (ResNet32-large variant).

*Comment on quaternionization.* Regarding the process of "quaternionization", i.e. creating an architecture that can be perceived as analogous or comparable to a real-valued one, we follow previous work (Parcollet et al., 2020; Sfikas et al., 2022; Comminiello et al., 2024; Parcollet et al., 2019b;a) and use the following trivial mappings from the real-valued to the quaternion domain, back and forth. In particular, assuming an input tensor is sized $\mathbb{R}^{H \times W \times C}$, we define a mapping $\mathcal{H} : \mathbb{R}^{H \times W \times C} \to \mathbb{H}^{H \times W \times C/4}$, by keeping coordinates $H, W$ identical in input and output, and rearranging input channels to channels, real and imaginary components of the output. In this manner, channel $c_r = 4c_h + b$ of the input, where $c_h \in [0..C/4 - 1]$ and $b \in [0..3]$ of the real-valued input is mapped to channel $c_h$ of the output, and the remainder is used to specify the

exact target channel. By convention, if $b = 0$ the input is mapped to the real component, and if $b = 1, 2, 3$ to imaginary components $\boldsymbol{i}, \boldsymbol{j}, \boldsymbol{k}$ respectively (Of course, numerous other conventions may be envisaged, mutually equivalent in practice). This may be a simple enough mapping, however it requires the number of channels to be a multiple of 4. Typically, the *input* and the *output* layer may pose a constrain in this regard: the input may be represented in a dimensionality that is not a multiple of 4 (e.g., color images have a dimensionality equal to 3), and the output may need to be of a dimensionality that is, again, not a multiple of 4 (e.g. number of classes on a classification task). In the literature, this has been worked around in a number of ways, out of which the most usual ones are either a) zero-padding the input (or output) with extra channels until they are a multiple of 4 b) use a $1 \times 1$ convolution that acts as a linear combination over channels or c) have real-valued convolution for the input (or output). In our implementation, we opt for the latter, to avoid introducing an unnecessary bias on the workflow in terms of comparing the quaternion-valued to the real-valued versions of the network. Importantly, the discussed mapping $\mathcal{H}$ is also a bijection, with an inverse $\mathcal{H}^{-1} : \mathbb{H}^{H \times W \times C/4} \to \mathbb{R}^{H \times W \times C}$. This mapping is defined in a completely analogous manner to $\mathcal{H}$, with real or imaginary channels mapped to the pseudoindex $b \in [0..3]$, and the same formula $(c_r = 4c_h + b)$ is used to map to real-valued indices from quaternion domain indices.

*Training protocol.* Models are trained for 80 epochs with SGD on cross-entropy loss (initial learning rate 0.1 - momentum 0.8) and mini-batches of 128. The learning rate follows a multi-step schedule, decayed by a factor of 0.1 at 50% and 75% of training.

