# OpenReview forum: "Unlocking the matrix form of the Quaternion Fourier Transform and Quaternion Convolution: Properties, connections, and application to Lipschitz constant bounding"
_TMLR — Accepted by TMLR_

### Review · Reviewer_QEr5 · 2025-08-26

**Summary Of Contributions:**

The authors investigate the connection between quaternionic circulant matrices and quaternionic Fourier matrices. They then use these results to efficiently constrain the eigenvalues of the quaternionic operators in a quaternion convolutional neural network. This serves as a regularization form that results on improved result using two ResNet-style model sizes on CIFAR10. Computing this projection without the presented results would generally be prohibitive.

**Audience:**

Yes

**Audience Explanation:**

Although not widely used, quaternion-based neural networks and advances towards making them usable in practice have some potential that may make them interesting to the community.

**Broader Impact Concerns:**

Some discussion about computational requirements would be welcome.

**Claims And Evidence:**

No

**Claims Explanation:**

Disclaimer: I am far from an expert in the field. I think I have been assigned this paper due to my past experience with rotation equivariant CNNs, which is unrelated to the work presented in this paper. I thus have a very superficial understanding of the topic of this paper and I am unable to assess the correctness of the theoretical results.
I will, therefore, focus on evaluating the experimental setup and results, and leave to more knowledgeable reviewers the evaluation of the correctness and relevance of the theoretical results.

1. I appreciate that the authors provide results for multiple random seeds in Fig 3. However, I would suggest rather showing these results by simply adding standard deviations to Tables 2 and 3. This would allow to also see the variation in the baselines.
2. There is an important lack of details about the experimental setup. I could not find what optimizer was used nor what hyperparameters. I can imagine the results may be different when different optimizers are used.
3. Although clipping the eigenvalues of the operators seems to improve results, there is no discussion about the alternatives (or lack thereof) to this. In real-valued neural networks, weight decay is generally used to regularize the model weights. A short discussion about whether it is possible or not to use a similar approach with QCNNs would be useful for readers like me.
4. I miss some comparisons to real-valued models, just as a reference, in terms of performance and running time.
5. The experiments only include one dataset (CIFAR10) and one architecture (ResNet, even if in two different sizes are used). This undermines the claims about the usefulness of the clipping approach made possible by the results in the paper. I understand there may be computational concerns regarding the extension of experiments, but there are small datasets that could be used (e.g. CUB), and architectures of similar size (VGG or ConvNext). Such extended experiments would be required to sustain the claim about usefulness of the clipping approach.
6. Other than the proof-of-concept in section 4, I would expect to see a discussion section about other possible applications of the paper’s results and QCNNs in general. In terms of real-valued models, CNNs tend to be replaced by non-(explicitly)-convolutional methods, such as those based on transformers or mamba. Any connection between the paper’s results (or related future work) and these other architectures would be useful.

**Requested Changes:**

I would like to see a response to each of the raised points:
1. Adding std to the results, rather than showing several runs as plots.
2. Adding optimization details.
3. Adding some discussion about alternative regularizations for QCNNs and their equivalents in real-valued models.
4. Comparisons to real-valued models.
5. Adding at least one additional dataset (I suggest CUB or some other small but challenging dataset).
6. Add a discussion about other possible applications of the proposed theoretical results.

---

> ### Author Response · Authors · 2025-09-15
> **Thank you for your feedback**
>
> We thank the reviewer for their careful and thoughtful feedback. In what follows, we begin addressing your comments. We will also upload changes that correspond to our responses in revised versions of the manuscript. These changes will be shown in blue for your convenience.

---

> ### Author Response · Authors · 2025-09-15
> **Connections to non-explicitly-convolutional operations**
>
> *Concerning your comment especially: “ In terms of real-valued models, CNNs tend to be replaced by non-(explicitly)-convolutional methods, such as those based on transformers or mamba. Any connection between the paper’s results (or related future work) and these other architectures would be useful.”*
>
> We agree that this is a very strong and interesting line of inquiry. Even though non-convolutional operations lie outside the focus of the paper’s contribution – the paper is primarily about the relation of Quaternion Convolution and QFT – due to the ubiquity of the convolution operation itself we can potentially envisage an impact that goes beyond the traditional schema of vanilla CNNs. It is true that there is a lot of work, especially in the context of neural nets, that is based on Transformers or the more recent mamba that seem to supersede or bypass the need for more “traditional” models like convolutional or recurrent nets. However, here we can begin with two broad lines of “defense” for convolution:
>
> * These architectures are too data hungry to be used efficiently in a great number of contexts. Some tasks must objectively be treated are resource-constrained, which means that using a larger-scale model does not translate well into tangible benefit. A good example here is maybe text recognition, a context in which transformers have indeed been used, but models that work only with self-attention are too big and too data-inefficient, while the improvement in efficiency that they provide is probably too little to be considered as to be worth their use.  [1] Concerning mamba, there is an interesting recent work from this year’s CVPR, questioning the need for mamba in vision [2].
>
> * Convolution, as a linear operation, can and has been used in tandem with layers of other types. A good example here is hybrid architectures that have emerged relatively recently. Models like the Convolutional Vision Transformer (CvT) replace the linear projections for queries, keys, and values with convolutional operations. Other approaches, such as the Conformer for speech recognition, place a dedicated convolution module within the Transformer block to capture local correlations alongside the self-attention mechanism's global context. Concerning Mamba: The Mamba architecture also includes a convolution layer as part of its internal block to extract local patterns before the "Structure State Space Model" focuses on long-range dependencies. There are also more direct fusions / hybrids with convnets like ConvMamba or VisionMamba. Graph-Convolution is another good example in the same rationale. Hence, in principle the results that we propose in this paper – and especially the proof-of-concept Lipschitz constant bounding method—can in principle be applied in Quaternionic versions of these models.  [3,4]
>
> In a somewhat related manner to (b) especially, we believe that it is noteworthy that 	convolution is intricately related to invariance/equivariance properties. For example, translational equivariance is related to standard convolution, and a line of research involves relating different kinds of equivariance to other forms of group convolution . [5]  (For us, this intricate relation with equivariance is probably the best argument that convolution is not going to become irrelevant any time soon). Concerning application of our results – in the form of the showcased bounding method or possibly another form --we think that this and the aforementioned examples can be good lines of research in perspective.
>
>
> As a disclaimer, let us stress again though: Our point of view in this paper is not about “proving” that convolution or quaternion convolution is “better” than other, non-convolutional operations. (Actually we would not ever agree with such a position – convolution is an extremely useful operation, as is self-attention, or non-linear operations and so on and so forth).
>
>
> [1] Wick et al., “Rescoring sequence-to-sequence models for text line recognition with CTC-prefixes", DAS 2022
>
> [2] “MambaOut: Do We Really Need Mamba for Vision?”, CVPR 2025
>
> [3] Wu et al., “Cvt: Introducing convolutions to vision transformers”, ICCV 2021
>
> [4] Chen et al., “Simple and deep graph convolutional networks”, ICML 2020
>
> [5] Kondor & Trivedi, “On the Generalization of Equivariance and Convolution in Neural Networks to the Action of Compact Groups”, ICML 2018

---

> ### Author Response · Authors · 2025-09-15
>
> We would love to hear your thoughts on this. If you think that part or all of the previous discussion makes sense and could or should be arranged in some form as part of the paper, we will gladly proceed with its integration.
>
> Thank you again for the feedback -- we will follow up with additional comments on other remarks in your review, as soon as possible.

---

> > ### Comment · Reviewer_QEr5 · 2025-09-16
> >
> > I am under the impression that motivating the choice of the research direction and the necessary background is important. I appreciate the new section 2, which will certainly be useful to readers not familiar with quaternions. In the same way, I was wondering if a section about the convolution operator would also make sense. Such a section could introduce the nature of the convolution operator and its relevance, where parts of the reflection in this response would fit.

---

> > > ### Author Response · Authors · 2025-09-21
> > > **Convolution**
> > >
> > > Thank you for the feedback! Indeed, a first mention of convolution on the preliminaries Section is useful. We have now included a paragraph referencing quaternionic convolution in Section 2.

---

> ### Author Response · Authors · 2025-09-17
> **Adding optimization details.**
>
> *There is an important lack of details about the experimental setup. I could not find what optimizer was used nor what hyperparameters. [..] Adding optimization details.*
>
> Here are details about the model and optimization used for the CIFAR-10 experiments. We have added this information as a separate Appendix.
>
> **Datasets and preprocessing**
>
> All initial experiments are conducted on CIFAR-10. Training images are augmented with random horizontal flips and random 32×32 crops after 4-pixel zero-padding. Inputs are converted to tensors and normalized channel-wise using the widely-adopted ImageNet statistics. Evaluation uses the non-augmented (only channel-wise normalization) pipeline.
>
> **Model**
>
> We use a CIFAR-style ResNet-32 topology with three stages of residual blocks and increasing channel width. Each stage has 5 residual blocks. Each block applies a 3×3 convolution, batch normalization, and ReLU; downsampling between stages is performed by strided convolutions. The network ends with global average pooling and a linear classifier. We then “quaternionize” the model by replacing the typical 3×3 convolution on the residual blocks with a depthwise quaternionic one. The input convolution layer is not replaced by a quaternionic one. To re-enable (non-quaternionic) channel mixing, we have a variant where a pointwise real convolution follows each introduced quaternionic depthwise convolution at the cost of more parameters (ResNet32-large variant).
>
> **Training protocol**
>
> Models are trained for 80 epochs with SGD on cross-entropy loss (initial learning rate 0.1 -  momentum 0.8) and mini-batches of 128. The learning rate follows a multi-step schedule, decayed by a factor of 0.1 at 50% and 75% of training.

---

> ### Author Response · Authors · 2025-09-17
> **Adding std to the results, rather than showing several runs as plots.**
>
> *Adding std to the results, rather than showing several runs as plots.[..] I appreciate that the authors provide results for multiple random seeds in Fig 3. However, I would suggest rather showing these results by simply adding standard deviations to Tables 2 and 3. This would allow to also see the variation in the baselines.[...]*
>
> *Query by Action editor: Personally, I would include both the current version and such an improved version, run on more than the current number (4) of random seeds (try 100 if possible, otherwise at least 10). This is because looking at the individual seeds do show that there is significant deviation from the mean which may be of a fat-tail type rather than something which can be captured in the variance.*
>
> We have re-run our experiments and modified Figure 3 to show mean and standard deviation as violin plots – we will shortly comply with showing the extra information you suggest in Tables 2 and 3. We have enriched the figure by showing two alternative options for the “c” parameter for clipping. We have used 20 random seeds for the experiments.

---

> > ### Author Response · Authors · 2025-09-18
> > **Adding std to the results, rather than showing several runs as plots. (contd.)**
> >
> > *However, I would suggest rather showing these results by simply adding standard deviations to Tables 2 and 3. This would allow to also see the variation in the baselines.*
> >
> > We have also added standard deviations to Tables 2 and 3, as well as in the plots of Figure 1.

---

> ### Author Response · Authors · 2025-09-21
> **Testing on additional datasets & backbone**
>
> *Adding at least one additional dataset (I suggest CUB or some other small but challenging dataset). [..] The experiments only include one dataset (CIFAR10) and one architecture (ResNet, even if in two different sizes are used). This undermines the claims about the usefulness of the clipping approach made possible by the results in the paper. I understand there may be computational concerns regarding the extension of experiments, but there are small datasets that could be used (e.g. CUB), and architectures of similar size (VGG or ConvNext). Such extended experiments would be required to sustain the claim about usefulness of the clipping approach.*
>
> Thank you for the suggestion. We have run experiments on additional datasets and an additional architecture, as the reviewer suggests. In particular, we have run tests on Fashion MNIST and CIFAR100, using the small and large Resnet variants we used in the paper. Furthermore, we have added another architecture for our tests, namely an adapted version of VGG16. In both cases we have run versions using clipping vs without clipping, like done in our previous CIFAR10 tests. Note also the low variance of reported accuracy w.r.t. different runs, which highlights the stability of our method. We have added a small section in the main text to introduce these results. We can clearly see that a similar trend is visible in our results as with our previous results: the method enabled by our theoretical results leads to direct gains in performance.
>
> We need to underline though, that we think that the main value of our paper is not in the quality of the Lipschitz constant bounding application (even though we have already discussed the merits of the method in terms of qualitative and quantitative results) -- this is just a “Proof-of-concept", in the sense that our main results (Section 3) make sense in at least one practical setting.

---

> ### Author Response · Authors · 2025-09-21
> **Adding some discussion about alternative regularizations for QCNNs and their equivalents in real-valued models**
>
> *"Adding some discussion about alternative regularizations for QCNNs and their equivalents in real-valued models. [..] Although clipping the eigenvalues of the operators seems to improve results, there is no discussion about the alternatives (or lack thereof) to this. In real-valued neural networks, weight decay is generally used to regularize the model weights. A short discussion about whether it is possible or not to use a similar approach with QCNNs would be useful for readers like me."*
>
> Thank you very much for the comment. We agree that regularization is a broad field with many effective strategies. Concerning comparison to other regularization strategies, our method effectively allows to efficiently compute the singular values of a convolutional operator, and stands as an adaptation of a form of Lipschitz regularization that has been extensively used in the real-valued domain [e.g. Sedghi et al. ICLR 2018, Grishina et al. ECCV 2024]. As pointed out by reviewer pLGg, Lipschitz constant bounding has uses other than improving generalization (a very important task in itself), like applications in Wasserstein GANs or Deep Equilibrium Networks. E.g. in WGANs, the stability and theoretical guarantees rely on the critic network being a 1-Lipschitz function.
>
> This stands in contrast to regularizers like L2 weight decay, which penalize the magnitude of weights without directly controlling the network's output sensitivity to input perturbations. While weight decay is a staple in real-valued neural networks, and applying it to quaternion NN learning should be straightforward (computing a quaternion norm should probably not require very important considerations for the quaternion domain), Lipschitz-based approach offers direct control over the network's smoothness, and finds use in the aforementioned domains (generatlization error control, WGANs, etc.)
>
> But are there alternatives to what we do in Section 4 with Lipschitz bounding, specifically targeting Quaternionic Convolution? Works discussing power methods in the context of quaternion matrices do exist, but as far as we know they either focus on the right spectrum (while we show that we need the left eigenvalues / the right spectrum) e.g. [Li et al, “On the power method for quaternion right eigenvalue problem”, Journal of Computational and Applied Mathematics (JCAM) 2019] or make assumptions about the matrix being Hermitian [Jia et al., “A new structure-preserving method for quaternion Hermitian eigenvalue problems”, JCAM 2013] and, more importantly, they do not take into account the intricate structure of the matrix that is induced by quaternionic convolution e.g. [Grasucci et al. “Quaternion Generative Adversarial Networks”, 2021]. The operator A of quaternionic convolution is of course linear, hence a study about Lipschitz constant bounding can focus on A, but we have discussed in the paper that this matrix can very quickly become too big to even fit in memory. While there are methods that focus on this for real-valued convolution, notably [Sedghi et al., “The singular values of convolutional layers”, ICLR 2018] (which is an inspiration for much of Section 4) or the much more recent [Grishina et al., “Tight and Efficient Upper Bound on Spectral Norm of Convolutional Layers”, ECCV 2024], the field regarding adaptations or novel methods in the quaternionic domain is rather poor. Hence, to the best of our knowledge, straightforward adaptations of power methods have not been proposed in past literature [Altamirano-Gomez et al., “Quaternion Convolutional Neural Networks: Current advances and Future Directions”, 2024].
>
> Furthermore, a naive implementation would have to solve fundamental issues in the quaternionic domain (but non-existent in real or complex domains), like taking into account that there are left and right spectra for each matrix, and these are completely (?) unrelated between themselves.
>
> Of course, this does not mean that better methods are not possible. On the contrary, we believe that the field is filled with interesting directions of research, like the one concerning power methods, implied by your comment, or adapting methods built on different rationales (see e.g. Table 1 of Grishina et al., ECCV 2024].
>
> Again, thank you very much for  the comment, and the suggestion of including a short related discussion. We will update the beginning of Section 4 with a relevant remark.

---

> ### Author Response · Authors · 2025-09-21
> **Extension of Section 5 with other possible applications of the paper's results**
>
> *Other than the proof-of-concept in section 4, I would expect to see a discussion section about other possible applications of the paper’s results and QCNNs in general. In terms of real-valued models, CNNs tend to be replaced by non-(explicitly)-convolutional methods, such as those based on transformers or mamba. Any connection between the paper’s results (or related future work) and these other architectures would be useful.*
>
> We have proceeded to extend Section 5 with two more specific examples of perspectives, to a large degree inspired by our discussion on "non-(explicitly)-convolutional methods" (see thread above). Thanks again and let us know for more suggestions!

---

### Review · Reviewer_pLGg · 2025-08-30

**Summary Of Contributions:**

The paper presents a new series of theoretical results regarding the singular values of circulant quaternionic matrices and their relationship to the quaternionic Fourier transform, extending popular results in the complex domain that relate the singular values of a circulant matrix to the Fourier transform of the associated filter.

The paper leverages these results to propose a new algorithm for efficiently computing the singular values of quaternionic (doubly) block-circulant matrices, which has a significantly smaller complexity than using the quaternionic version of the singular value decomposition (ie without exploiting the block-circulant structure).

Throughout a series of CIFAR10 classification experiments with quaternionic ResNets, the efficient singular value algorithm is used to bound the Lipschitz constant of quaternionic neural networks in a reasonable amount of time, in order to improve the generalization of such networks.

**Audience:**

Yes

**Audience Explanation:**

I believe this paper could be of interest to machine learning researchers developing quaternionic neural networks and other algorithms leveraging quaternions.

**Broader Impact Concerns:**

I don't have any ethical concerns regarding this work, which is mostly theoretical.

**Claims And Evidence:**

Yes

**Claims Explanation:**

All theoretical results in the paper are supported by proofs, which appear correct to me, although I did not fully verify their correctness, as I'm not an expert on quaternions. The experimental results are convincing as well, demonstrating clear evidence for the impact of spectral norm clipping on the performance of quaternionic neural networks. My only concern regarding these results is the (I believe) unfair comparison with real counterparts: I believe that the authors should compare with non-depth-wise convolutions that can mix channels, as quaternionic networks do mix channels due to the Hamilton product. The current comparison is slightly misleading, as it leads to the conclusion that quaternionic networks are much better than their real counterparts by choosing an unrealistic backbone architecture.

**Requested Changes:**

I believe that the presentation in the paper could be greatly improved by polishing the mathematical notation, providing an introduction to quaternions for non-experts, fixing typos and flawed sentences, and moving some technical lemmas/proofs to the appendix.

I also have a concern regarding computing the spectral norm: why not use power methods, which are generally much less expensive than computing all singular values? Is there a hidden assumption about the non-linearity of the network? (I guess some non-linearities could have large Lipschitz constants, and break the constant of the overall network)
It would also be good to mention other applications of training networks with a constrained Lipschitz constant beyond improving generalization, such as Wasserstein GANs or convergent plug-and-play/deep equilibrium networks.

Notation:
- I would use ^\top instead of ^T for transposing
- I would explain what \mathbb{H} is at the beginning of the paper (no more than half a page) and briefly introduce basic concepts around quaternions, ie what is a unit quaternion, how do we write a quaternion etc.
- I find using uppercase for the Fourier transform of a vector confusing, I would keep any vector lowercase
- I would write "j=0..N" -> "j=0,1,\dots,N"
- I would use begin{proof} end{proof} for proofs. For example, the beginning of the proof of prop. 3.3 is unclear
- modulo-N -> modulo-$N$
- for norms I would use the latex command "\|" instead of "||" (openreview doesn't display it well, but it should give two bars on latex)
- I would use $\ell$ instead of $l$
- I would use \text{} for rank(.), arccos(.), qsvd(.) etc


Presentation:
- I would move some proofs and technical lemmas to the appendix (for example, lemmas 4.1 and 4.2) - I find the current organization quite dense to read, hindering a global understanding of contributions/ideas.
- The definition of a layer could be improved at the beginning of page 11. The term "linearity B" could be simplified to the "linear layer" or "matrix" B.
- Table 1 could be improved, currently it is hard to understand from the caption what is the difference between SV and Full, and its confusing that the time unit is reported in some cell "hours" whereas is not explained in others.

Typos/Flawed sentences:
- the first footnote appears after the full stop, instead of before. There's also some unnecessary blank space before footnotes
- page 2, remove "In our opinion, it is notable that "
- page 2, contribution 3 could be simplified, it is confusing to refer to "the previous result" where no result has been shown so far.
- page 3, "quternion", "wherever"
- page 5, blank space missing "Notation Conventions)For columns"
- page 14, "Times to just" -> "Computational times
- page 15 "c = 1.0." -> "c=1."
- Table 1: "2D kernel side size" is confusing
- page 16, I'm not sure what "To this end we continue out experimentation with the more “interesting" case of the small version." means

---

> ### Author Response · Authors · 2025-09-14
> **Thank you for your feedback**
>
> Thank you so much for all the effort you put to read and review our work. In what follows, we (begin to) provide remarks on the comments you note. We will also upload changes that correspond to our responses in revised versions of the manuscript. These changes will be shown in blue for your convenience.

---

> ### Author Response · Authors · 2025-09-14
> **Moving lemmas and other text to the appendix**
>
> *I believe that the presentation in the paper could be greatly improved by polishing the mathematical notation, providing an introduction to quaternions for non-experts, fixing typos and flawed sentences, and moving some technical lemmas/proofs to the appendix. (related requested change: I would move some proofs and technical lemmas to the appendix (for example, lemmas 4.1 and 4.2) - I find the current organization quite dense to read, hindering a global understanding of contributions/ideas.)*
>
> We agree that the text is overall dense, and refactoring and repositioning could be helpful. We have moved proofs to lemmas 4.1 and 4.2 to the appendix. We have also moved corollary 4.4.1 to the appendix. We hope that this makes the text easier to read through. Please let us know if you think this is enough, or if you think that moving more material to the end would be beneficial in terms of conciseness and clarity.

---

> ### Author Response · Authors · 2025-09-14
> **Moving introductory material to the main text**
>
> *[..] providing an introduction to quaternions for non-experts [..]*
>
> We have also carried out the refactoring you suggested regarding the introduction to quaternions (another reviewer, 4wny, has also suggested the same modification). We agree that, especially regarding quaternionic matrices, this text should include preliminaries as part of the main text. This is now followed with what was previously Section 2, in order to make the connection smoother with current practices and challenges, and Section 3, which contains the main results. Please feel free to tell us if you agree with the specific way we carried out this refactoring. In particular, tell use if you think that more (or less) material should be moved to the intro, or changed / modified in any way.

---

> ### Author Response · Authors · 2025-09-14
> **Improvements in presentation of Tables and change of term "linearity"**
>
> *Table 1 could be improved, currently it is hard to understand from the caption what is the difference between SV and Full, and its confusing that the time unit is reported in some cell "hours" whereas is not explained in others.*
>
> We agree that Table 1 should be improved. We have replaced the text “hours”, made the caption clearer, and changed the layout of the whole table. Furthermore, we have homogenized Table appearance to have captions before content in all cases.
>
> *The definition of a layer could be improved at the beginning of page 11. The term "linearity B" could be simplified to the "linear layer" or "matrix" B.*
>
> We have changed the term “linearity” to “linear layer” in both instances.

---

> ### Author Response · Authors · 2025-09-14
>
> Thanks again for the feedback. We will follow up with additional comments on other remarks in your review -- especially regarding comparisons vs real-valued nets and other concerns.

---

> ### Author Response · Authors · 2025-09-18
> **Polishing the mathematical notation, fixing typos and flawed sentences**
>
> *I believe that the presentation in the paper could be greatly improved by polishing the mathematical notation, providing an introduction to quaternions for non-experts, fixing typos and flawed sentences, and moving some technical lemmas/proofs to the appendix.*
>
> Thank you for pointing out these changes and these errata. We have carried out the proposed changes. Here are some comments on part of them and/or comments on some additional changes, and a short explanation on why we have counter-suggested to have some exceptions:
>
> * Also corrected some instances of [1,N] to [0,N-1] for notational coherency (e.g. in Proposition 3.1).
> * Corrected: with $\lambda^\mu \subset \sigma_l(C)$, On corollary 3.6.1 . This way it is clearer that we are discussing the *left* spectrum here.
> * “I find using uppercase for the Fourier transform of a vector confusing, I would keep any vector lowercase” . We completely understand your point, and we normally don't use capital notation for vectors; however, we think that if we used a different letter or a variation of the letter it would have been much more confusing than the current choice.
> * *We have replaced || with just a single bar where it was more appropriate (Proposition 3.7), and with \| anywhere else (eigenvectors, other vectors).* We have replaced l with \ell . We used \text with arccos, rank, qsvd.
> * *“the first footnote appears after the full stop, instead of before. There's also some unnecessary blank space before footnotes”.* We corrected this. Also, we used “\!” to trim the unnecessary blank space a bit.
> *“page 16, I'm not sure what "To this end we continue out experimentation with the more “interesting" case of the small version." Means"* Thank you for pointing this out – indeed we think that this part of the text should really be refactored. We agree that this particular sentence looks somewhat cryptic. It was due to an attempt to maybe overexplain things from our side. The point here is that we continue experiments on the “small” version, since we find that in both versions we have an analogous behaviour in favor of clipping. We have deleted the comment about sensitivity of the large model, as we find that it indeed is confusing and does not contribute to anything in the narrative anyway.
> * We have already move part of the proofs to the appendix (see previous comment)

---

> ### Author Response · Authors · 2025-09-21
> **Concerning other uses of the Lipschitz constant (determination and bounding)**
>
> *It would also be good to mention other applications of training networks with a constrained Lipschitz constant beyond improving generalization, such as Wasserstein GANs or convergent plug-and-play/deep equilibrium networks.*
>
> We are grateful for this very useful remark, as this will markedly improve the perceived scope of the proposed proof-of-concept application. We have included this text in the beginning of Section 4: “Bounding the Lipschitz constant has been shown to be a way to control the generalization error (Prince, 2023, Chapter 9, Appendix B). Determination or bounding of the constant also ﬁnds other important uses, which include application in Wasserstein GANs Petzka et al. (2018), (Prince, 2023, Chapter 15) or plug-and-play networks Ryu et al. (2019) and deep equilibrium (DEQ) networks Pabbaraju et al. (2021). “. Also this text in the conclusion: “Lipschitz constant bounding and determination is important in a number of important applications, and important in diﬀerent classes of models, including Wasserstein GANs Petzka et al. (2018), Plug-n-Play networks Ryu et al. (2019), or Deep Equilibrium Networks Pabbaraju et al. (2021); hence, one direct line of research involving our results could envisage generalizations of these applications in quaternionized versions of these models.”

---

> ### Author Response · Authors · 2025-09-21
> **Channel mixing in Quaternionic NNs vs real-valued counterparts**
>
> *My only concern regarding these results is the (I believe) unfair comparison with real counterparts: I believe that the authors should compare with non-depth-wise convolutions that can mix channels,  as quaternionic networks do mix channels due to the Hamilton product. The current comparison is slightly misleading, as it leads to the conclusion that quaternionic networks are much better than their real counterparts by choosing an unrealistic backbone architecture.*
>
> Thank you for this point of clarification. It allows us to better articulate the scope and intention of our work.
> We do not intend to present quaternionic networks as being generally superior to their real-valued counterparts. Instead, the primary objective of our paper is to provide a theoretical framework (Section 3) and a practical tool for a specific class of models. The experimental results in Section 4 are intended as a "proof-of-concept," demonstrating that our theoretical findings have a valuable application within the context of QNNs.
>
> As the reviewer correctly notes, a direct comparison of QNNs with real-valued models must account for architectural differences. We have previously highlighted this distinction in the manuscript, noting that "cascades of depthwise convolutions in R are much less expressive than their counterpart in H, because the former represent series of convolutions over a single real-valued channel. The latter however, do comprise a form of “channel mixing” under the hood, due to the deﬁnition of the Hamilton product."
> So, we do not hide the fact that depthwise convolutions in H do have the trait that you correctly note – inherently they do channel mixing due to the Hamilton product.
>
> In a nutshell, our intention with this work is not to show that quaternionic networks are in general / universally better in all contexts, compared to their real-valued counterparts. On the contrary,  Section 4 acts as a “Proof-of-concept" that our results of Section 3 do have a practical application – this application is in the context of Quaternion NNs. So, given that we want to work with Quaternion NNs, we now have an extra set of results that can act as valuable tools, plus a very specific Lipschitz bounding method. We do believe that Quaternion NNs, and more broadly, quaternionic representation, have important merits, which we have tried to outline in Section 2 (now subsection of the current Section 2).

---

> ### Author Response · Authors · 2025-09-21
> **Concerning alternatives to computing the spectral norm**
>
> *"I also have a concern regarding computing the spectral norm: why not use power methods, which are generally much less expensive than computing all singular values? Is there a hidden assumption about the non-linearity of the network? (I guess some non-linearities could have large Lipschitz constants, and break the constant of the overall network)."*
>
> Thank you for the question. Before proceeding with the answer in itself, let us note what is for us an important “disclaimer”: The method in Section 4 acts in this paper as a proof of concept application of our results. This does not mean that it is not useful – on the contrary, we show that if we need to obtain the eigenstructure of a quaternionic convolutional operator, it is the best thing we can do. This is to be taken in the sense of an example application of the usefulness of what we proved.
>
> Concerning the specifics of the (otherwise very valid) concern raised by the reviewer: Let us clarify that computation of singular values under the method described in Section 4 happens in a single iteration, in the sense of not requiring to be re-run for each singular value required separately. In other words, a single QFT is enough to obtain all left eigenvalues in one go (there are N QSVDs after this step but they are all extremely cheap, as at worst each is applied on a 2x2 matrix). So computing all singular values vs one value does not bring any overhead at all. To address the reviewer's question on non-linearity, our method makes no hidden assumptions -- The Lipschitz constant is bounded by the product of the Lipschitz constants of its individual layers, and for a linear layer, the Lipschitz constant is its spectral norm.
>
> But are there alternatives to what we do in Section 4 with Lipschitz bounding, specifically targeting Quaternionic Convolution? Works discussing power methods in the context of quaternion matrices do exist, but as far as we know they either focus on the right spectrum (while we show that we need the left eigenvalues / the right spectrum) e.g. [Li et al, “On the power method for quaternion right eigenvalue problem”, Journal of Computational and Applied Mathematics (JCAM) 2019] or make assumptions about the matrix being Hermitian [Jia et al., “A new structure-preserving method for quaternion Hermitian eigenvalue problems”, JCAM 2013] and, more importantly, they do not take into account the intricate structure of the matrix that is induced by quaternionic convolution e.g. [Grasucci et al. “Quaternion Generative Adversarial Networks”, 2021]. The operator A of quaternionic convolution is of course linear, hence a study about Lipschitz constant bounding can focus on A, but we have discussed in the paper that this matrix can very quickly become too big to even fit in memory. While there are methods that focus on this for real-valued convolution, notably [Sedghi et al., “The singular values of convolutional layers”, ICLR 2018] (which we do reference in our paper) or the much more recent [Grishina et al., “Tight and Efficient Upper Bound on Spectral Norm of Convolutional Layers”, ECCV 2024], the field regarding adaptations or novel methods in the quaternionic domain is rather poor. Hence, to the best of our knowledge, straightforward adaptations of power methods have not been proposed in past literature [Altamirano-Gomez et al., “Quaternion Convolutional Neural Networks: Current advances and Future Directions”, 2024].
>
> Furthermore, a naive implementation would have to solve fundamental issues in the quaternionic domain (but non-existent in real or complex domains), like taking into account that there are left and right spectra for each matrix, and these are completely (?) unrelated between themselves.
>
> Of course, this does not mean that better methods are not possible. On the contrary, we believe that the field is filled with interesting directions of research, like the one concerning power methods, implied by your comment, or adapting methods built on different rationales (see e.g. Table 1 of Grishina et al., ECCV 2024].
>
> We have already adapted the text to integrate your suggestion about referring to other important uses of Lipschitz constant bounding (see answer to comment above). If you think that we should make further changes in-text regarding this point, please free to make suggestions.

---

> > ### Comment · Reviewer_pLGg · 2025-09-26
> > **Thanks for addressing my concerns**
> >
> > Thank you for the detailed answers addressing my concerns.
> > I believe that the updated manuscript has clearer notation and better tables, and the explanations regarding the challenges of a quaternionic power method are convincing. I very much appreciated the introduction to quaternions.
> >
> > I have very minor final points that could be improved:
> > - Figures 2 and 4 could be included in .pdf format to improve their resolution (I would personally reduce the size of the arrows, but this is a purely stylistic opinion)
> > - "observational models" -> "observation models"
> > - exp in page 24 lacks the \
> > - some references on page 24 are wrongly formatted (use \citep instead of \cite)

---

> > > ### Author Response · Authors · 2025-09-30
> > > **Thanks!**
> > >
> > > Thank you so much! We have updated the document following all your last suggestions&corrections as well.

---

> ### Comment · Action_Editor_dze6 · 2025-09-26
> **Follow up**
>
> Dear Reviewer,
>
> As all reviewers will have to submit their recommendations in the near-ish future, I think it would be good if you could interact with the authors.
>
> It appears the main concerns you raised are related to the clarity of the writing, which I shared based on the original version of the manuscript. The authors have attempted to fix all the issues and submitted a revision. Could you take a look at it, together with the responses by the authors, and see whether this has clarified your doubts?
>
> Looking forward to hearing back from you soon,
>
> Best regards,
> AE

---

### Review · Reviewer_4wny · 2025-09-08

**Summary Of Contributions:**

This work studies the matrix form of the Quaternion Fourier Transform (QFT) and quaternionic convolutions. It focuses on showing how the classical results can extend, in many cases with non-trivial ways, to the quaternionic domain. The paper first defines the family of Quaternionic Fourier matrices $Q_N^\mu$ (given a pure unit quaternion $\mu$), and shows how the columns of $Q_N^{-\mu}$ are eigenvectors of the quaternionic circulant $C$ matrices. These results,  which connect the Quaternionic Fourier matrices with the eigenvalues/eigenvectors of the quaternionic circulant matrices,  are then extended to the doubly block-circulant matrices, which allows the authors to handle cases of 2D convolution. Building on their results, the authors propose a method for computing and bounding the spectral norm of quaternionic convolutional layers without explicitly forming the large doubly-block circulant matrix that densely describes the 2D convolution. Finally, they validate the significant computational gains of their proposed method for bounding the spectral norm of quaternionic 2D convolution, in a ResNet-style architecture trained on CIFAR-10.

**Strengths**:
- The authors provide a clear formulation of the QFT matrices and their relationship to the DFT. They show that $Q_N^\mu$ generalizes the complex DFT matrix in the quaternionic domain, with the standard DFT being recovered when $\mu=i$.
- The extension of the results to the doubly block-circulant quaternionic matrices is quite helpful since it allows the results to be naturally applied in 2D convolution and all subsequent tasks that utilize them.
- Such a benefit is shown in the provided proof of concept application of bounding the spectral norm of quaternionic 2D convolutions.
- The appendix provides a nice overview of the properties of quaternions, making the work accessible to readers who are not familiar with the topic. However, I believe that some of these introductory definitions would be better located at the start of the main text to create a more linear reading flow.

**Weaknesses**:
- Some of the proofs contain mistakes/typos that make it harder to evaluate their correctness. For example, in equation (12), where is the definition of $h[i]_N$? Also, after equation (12), the authors refer to the $k[i]_N$ without using it. Are $h[i]_N$ and $k[i]_N$ the same thing, and the use of $k$ is a typo?  Similarly, in proposition 3.8, $\mu$ is defined but never used, while $x$ is used but never defined. While these mistakes might be small, they reduce the confidence of the reader who tries to understand the theoretical formulations.
- Proposition 3.8 is provided without any proof. For the completeness of the paper, all provided propositions must be accompanied by a proof, even if it is located in the appendix. Similarly some of the corollaries (3.6.1, 3.6.2, 3.7) are not accompanied by a proof, although in this case they might be considered too trivial to be included.

**Audience:**

Yes

**Audience Explanation:**

This work provides a significant contribution that can be of interest to the TMLR community since it performs an in-depth analysis of the properties of QFT matrices that can enable practitioners to incorporate quaternionic convolutions in machine learning models more easily.

**Broader Impact Concerns:**

There are no concerns about the ethical implications of the work that would require adding a Broader Impact Statement

**Claims And Evidence:**

Yes

**Claims Explanation:**

**Clarity**: Apart from the mistakes/typos noted in the weaknesses, the paper is well structured and easily readable even to an audience that might not be an expert on the topic.

**Correctness**: Overall, given that the authors address the mistakes/typos, all of the formulations and theoretical results appear sound. In addition to the proofs, the authors showcase how the above results provide actionable tools by describing a proof-of-concept application of spectral bounding of quaternionic convolution in Section 4.

**Requested Changes:**

**Critical changes**:
- The authors should address the two weaknesses regarding the typos in the proofs and the fact that some proposition corollaries are not accompanied by a proof. This is critical for the publication of this work since it affects the correctness of the presented results.

**Minor Changes and Suggestions**:
- In the proposed application of bounding the spectral norm of the quaternionic convolutions, it would be interesting to also study the numerical stability of the proposed method in comparison with the brute-force method. Such a comparison would be quite interesting since the spectral norm normalization methods are often numerically unstable when incorporated in a learning pipeline.
- One possible suggestion, mentioned also in the last bullet point of Strengths, is to include some of the preliminary definitions regarding the quaternions at the beginning of the main text, so that the paper has a more linear reading flow.

---

> ### Author Response · Authors · 2025-09-14
> **Thank you for your feedback**
>
> We deeply appreciate the time and effort for carrying out the review. We are very glad that you find that the paper is useful – in what follows, we begin addressing the comments you note. We will also upload changes that correspond to our responses in revised versions of the manuscript. These changes will be shown in blue for your convenience.

---

> > ### Author Response · Authors · 2025-09-14
> > **Regarding some typos/errata**
> >
> > *Some of the proofs contain mistakes/typos that make it harder to evaluate their correctness. For example, in equation (12), where is the definition of $h[i]_N$? [..]  Similarly, in proposition 3.8, μ is defined but never used, while x is used but never defined. While these mistakes might be small, they reduce the confidence of the reader who tries to understand the theoretical formulations.*
> >
> > Thank you for pointing this out. This is indeed a typo: k[i]_N should be h[i]_N, we have corrected this. We have also modified the text to better highlight the relation of h (which is a circulant matrix kernel) to C, which is the circulant matrix that h generates. Concerning Proposition 3.8, we have proceeded to write a complete and detailed proof (see answer that follows). We have also corrected notation and errata (k_C -> k_D, unused μ). μ is indeed never used, we apologize for this -- we have erased the related phrase. Concerning “x”, this is defined in the original text with the sentence: “The operator $vec(x)$ represents column-wise vectorization of signal $x \in \mathbb{H}^{M \times N}$.

---

> ### Author Response · Authors · 2025-09-14
> **Addition of proofs to proposition 3.8, and proofs to corollaries 3.6.1, 3.6.2**
>
> *Proposition 3.8 is provided without any proof. For the completeness of the paper, all provided propositions must be accompanied by a proof, even if it is located in the appendix. Similarly, some of the corollaries (3.6.1, 3.6.2, 3.7) are not accompanied by a proof, although in this case they might be considered too trivial to be included.*
>
> Thank you for the suggestion. We agree that this is important to show in more detail, as 2D convolution is evidently very important in a range of contexts, including foremost imaging. This proposition is analogous to Proposition 3.3(3), and aims to show the relevance of doubly-block circulant matrices and connection to 2D (Quaternion) Convolution. In the revised document, we have proceeded to include a full proof of this result. We will be happy to proceed with a more extended clarification if you feel that it is required.
>
> Concerning Corollary 3.6.1: This result follows from 3.6, plus the fact that that Q_N^{-μ} contains eigenvectors for any circulant matrix, due to Proposition 3.5. We have added in-text an explanation for this corollary, highlighting this point.
>
> Concerning Corollary 3.6.2: We have slightly refactored text here, moving a phrase as part of the proof, and extended with a short addition to the explanation. This diagonalization form means that AS = SΛ, but by the definition of quaternionic matrix multiplication (see e.g. Fuzhen Zhang 1997) this form corresponds to Ax = xλ, which defines right eigenvalues. For left eigenvalues we would require Ax = λx, which is not the case here. Rephrased as a proof by contradiction, by the above argument any diagonalization A = SΛS^{-1} gives us right eigenvalues, and obtaining left eigenvalues by such a form would mean that left and right eigenvalues coincide. This is known to not be the case in general (Fuzhen Zhang, 1997).

---

> ### Author Response · Authors · 2025-09-14
> **Other clarifications -- all claims now covered with a proof**
>
> *The authors should address the fact [..] that some proposition corollaries are not accompanied by a proof.*
>
> Aside from the corollaries noted above, Corollary 3.6.3 is the only one that has been left without proof, so we have proceeded to cover this as well. We have also added a short explanation for Corollary 3.6.3, clarified text to note that we are referencing QFT in eq. 11, and included a phrase to make the connection to the text that follows smoother. If you feel that a point in the above parts of the text (or elsewhere) should be made clearer, please do let us know.

---

> ### Author Response · Authors · 2025-09-14
> **Moving introductory content to the main text**
>
> *The appendix provides a nice overview of the properties of quaternions, making the work accessible to readers who are not familiar with the topic. However, I believe that some of these introductory definitions would be better located at the start of the main text to create a more linear reading flow.*
>
> As also requested by reviewer pLGg, we have carried out the requested change. We agree that moving part of the text from the appendix to the main text, in order to provide a smoother intro to the main material is a good idea. Please let us know if this change suits you and/or you think that other parts should make it into the main text / or be moved back. Likewise, feel free to tell us your opinion about having the previous Section 2 as a subsection in this version.

---

> ### Author Response · Authors · 2025-09-14
>
> Thanks again for the feedback. We will follow up with additional comments on other remarks in your review (especially concerning numerical stability).

---

> ### Author Response · Authors · 2025-09-21
> **Stability tests**
>
> *In the proposed application of bounding the spectral norm of the quaternionic convolutions, it would be interesting to also study the numerical stability of the proposed method in comparison with the brute-force method. Such a comparison would be quite interesting since the spectral norm normalization methods are often numerically unstable when incorporated in a learning pipeline.*
>
> Thank you for this suggestion. We have prepared a batch of tests where quaternionic convolution kernels of different sizes are iteratively clipped & reconstructed several times, after bouts of same-seed noise injection. To this end, we have added a paragraph explaining the experiment and the results, in Section 4. This shows that clearly the two methods have very similar results, and even if applied over a great number of times, there is no noticeable noise that would propagate due to numerical inefficiency.
>
> We had also initially prepared a visualization of kernel clipping with our method, where we visualized the quaternion field of the kernel via two ways: a) a colour image b) a set of spatial / 3D rotations, using the Mollweide projection-based method of [Murphy et al., “Implicit Representation of Probability Distributions on the Rotation Manifold”, 2021]. We weren’t sure about the usefulness of this however, so we left it out. You can take a glipse on this link: https://freeimage.host/i/KYEp7St, and if you think that it would be a good idea to eventually put it back in the revised paper, feel free to suggest so.

---

> ### Comment · Action_Editor_dze6 · 2025-09-26
>
> Dear Reviewer,
>
> Could you please read the authors' rebuttal and double check whether you are happy with the corrections?
>
> Best wishes,
>
> AE

---

> ### Comment · Reviewer_4wny · 2025-10-08
> **Thank you for the Rebuttal and Changes**
>
> I thank the authors for addressing my concerns and updating the manuscript. I appreciate the correction of the typos and the addition of the proofs that were missing in the initial submission.
> I believe the changes, especially the inclusion of more extensive preliminaries and additional figures, make the presentation of this work much more accessible to a broader audience.
> In its current state, this work provides a well-supported presentation of various results and tools on the Quaternionic Fourier Transform and Convolution that would be of great interest to the TMLR community.

---

### Review · Reviewer_3wXS · 2025-09-23

**Summary Of Contributions:**

1. Generalization of Matrix Representations to the Quaternion Domain: The paper establishes matrix forms for Quaternionic Convolution and the Quaternion Fourier Transform (QFT), demonstrating that they possess properties analogous to their real and complex counterparts. It reveals that the standard complex Fourier matrix is a special case within an infinite family of QFT matrices, which are interconnected via a rotation-like operation.

2. Elucidation of the Connection between Circulant and QFT Matrices: The work clarifies the nuanced relationship that links quaternionic circulant matrices (representing convolution) and QFT matrices in the quaternion domain. A notable finding is that the left eigenvalues of a convolution kernel are functions of both the left and right QFTs.

3. A Method for Left Eigenvalue Computation and Reconstruction: Addressing the challenge that the left spectrum of a quaternion matrix is generally difficult to compute, the paper introduces an efficient method to calculate a specific set of N left eigenvalues for an N×N circulant matrix using the QFT. It further proves that this set of eigenvalues is sufficient to uniquely reconstruct the original circulant matrix under specific conditions.

**Audience:**

Yes

**Audience Explanation:**

The paper bridges theory with a practical application in deep learning, making it appealing to several key groups within the machine learning research community.

**Broader Impact Concerns:**

There are no concerns about the ethical implications of the work that would require adding a Broader Impact Statement

**Claims And Evidence:**

Yes

**Claims Explanation:**

The core theoretical contributions presented in Section 3 of the paper, and  The authors conducted experiments to verify that their method.

**Requested Changes:**

The paper presents a valuable contribution by extending the theory of circulant matrices and the Fourier Transform to the quaternion domain and demonstrating its application for regularizing Quaternionic Neural Networks (QNNs). The theoretical proofs are sound, and the empirical results are convincing. However, the work could be made significantly more accessible and its impact broadened with a few adjustments focused on improving exposition and contextualizing the experimental results.

1. Enhancing Intuition for Theoretical Results. The paper is theoretically dense and jumps directly from propositions to formal proofs. This approach may be difficult for a broader machine learning audience to follow, especially for the non-intuitive results stemming from quaternion non-commutativity (e.g., the connection between left eigenvalues and the right QFT in Proposition 3.5). I suggest adding a few sentences or a short paragraph after key propositions (especially 3.4, 3.5, and 4.3) to provide a more high-level, intuitive explanation of the result.

2. Broadening the Introduction and Discussion. The work is primarily framed within the context of QNNs. While this is the direct application, the core contribution—an efficient method to analyze the spectral properties of a convolutional operator—has broader implications. I recommend expanding the introduction and conclusion to better situate the work within the wider field of deep learning. The authors could emphasize that they are solving a general problem of efficiently analyzing convolutional layers, and the quaternion domain is a particularly challenging and interesting instance of this problem.

3. Contextualizing Experimental Performance with Other Regularizers. The experiments convincingly show that applying spectral norm clipping (enabled by the paper's method) improves performance compared to not applying it. However, it is unclear how this method compares to other standard regularization techniques when applied to QNNs, such as weight decay or dropout.

4. Discussing Applicability to Modern Architectures. The experiments are conducted on VGG and ResNet-style architectures. While these are standard, the deep learning landscape is rapidly evolving with architectures like Vision Transformers (ViTs), ConvNeXts, and State Space Models (e.g., Mamba).

---

> ### Author Response · Authors · 2025-09-26
> **Enhancing Intuition for Theoretical Results.**
>
> *Enhancing Intuition for Theoretical Results. The paper is theoretically dense and jumps directly from propositions to formal proofs. This approach may be difficult for a broader machine learning audience to follow, especially for the non-intuitive results stemming from quaternion non-commutativity (e.g., the connection between left eigenvalues and the right QFT in Proposition 3.5). I suggest adding a few sentences or a short paragraph after key propositions (especially 3.4, 3.5, and 4.3) to provide a more high-level, intuitive explanation of the result.*
>
>
> Thank you for this remark. We have tried to answer this query by going a bit further and adding a number of figures in the text, to the end of helping break down the basic concepts in a high level. Here is a short list of our additions:
> * Figure 1: A figure that introduces the “protagonist” of the paper: The quaternionic matrix.
> * Figure 2: A quick intro into the intricacies of quaternionic eigenstructure.
> * Figure 3: The relation between the DFT and QFT. This relates especially to Propositions 3.3 and 3.4.
> * Figure 4: A visualization of our key results in Proposition 3.5.
> * Figure 5: A visual summary of the procedure outlined in Section 4 (proof of concept).
>
> We hope that his addition fits to the rationale of your suggestions. Please let us know whether this modification is compatible to your expectation, and whether you think that it is useful for a high-level understanding of the concepts.
>
> We will comply with an answer to the remainder of your comments as soon as possible. Thank you again for your suggestion.

---

> ### Author Response · Authors · 2025-09-30
> **Broadening the Introduction and Discussion & Applicability to modern architectures**
>
> *Broadening the Introduction and Discussion. The work is primarily framed within the context of QNNs. While this is the direct application, the core contribution—an efficient method to analyze the spectral properties of a convolutional operator—has broader implications. I recommend expanding the introduction and conclusion to better situate the work within the wider field of deep learning. The authors could emphasize that they are solving a general problem of efficiently analyzing convolutional layers, and the quaternion domain is a particularly challenging and interesting instance of this problem.*
> and *Discussing Applicability to Modern Architectures. The experiments are conducted on VGG and ResNet-style architectures. While these are standard, the deep learning landscape is rapidly evolving with architectures like Vision Transformers (ViTs), ConvNeXts, and State Space Models (e.g., Mamba).*
>
> We have added the following paragraph in the introduction: "From another standpoint, we can frame our work as serving to elucidate the properties of Convolution in H. We contribute to the better understanding of the convolutional operator, and provide the means to analyze
> and manipulate its spectral properties. Convolution is an ubiquitous operation in data science and machine
> learning, and its instance in the quaternion domain forms a challenging, yet interesting and promising tool
> that is hitherto underexplored. The extensive use and long history of convolution in the context of neural
> networks is very well-known (Prince, 2023); while it can be argued that the ﬁeld is rapidly evolving with
> non-convolutional backbones such as Transformers or State-Space models (Gu & Dao, 2023), the intrinsic
> connection of convolution with notions such as invariance and equivariance (Kondor & Trivedi, 2018) or
> the already known connection to frequency analysis in R (Jain, 1989) (and extended with the current work
> in H), are strong pointers in favor of the continued relevance of convolution. The success of hybrids of
> non-convolution with convolution or convolutional components seem to further corroborate this point (Wick
> et al., 2022; Zhang et al., 2019; Xu et al., 2023; Gu & Dao, 2023; Yu & Wang, 2025). Our work serves to
> further widen the usefulness and domain of application of convolution."
>
>
> We have also added text covering similar considerations in the conclusion, with a special reference in architectures such as transformers or mamba. We hope that this addition serves to address your points about framing our work in the context of the applications of convolution and deep learning. Please let us know if you feel that a modification or extension (or even that we have completely misunderstood your point?) is in order. Thank you again for the suggestion.

---

> ### Author Response · Authors · 2025-09-30
> **Concerning other regularizers**
>
> *Contextualizing Experimental Performance with Other Regularizers. The experiments convincingly show that applying spectral norm clipping (enabled by the paper's method) improves performance compared to not applying it. However, it is unclear how this method compares to other standard regularization techniques when applied to QNNs, such as weight decay or dropout.*
>
> Concerning comparison to other regularization strategies, our method effectively allows to efficiently compute the singular values of a convolutional operator, and stands as an adaptation of a form of Lipschitz regularization that has been extensively used in the real-valued domain [e.g. Sedghi et al. ICLR 2018, Grishina et al. ECCV 2024]. As pointed out by reviewer pLGg, Lipschitz constant bounding has uses other than improving generalization (a very important task in itself), like applications in Wasserstein GANs or Deep Equilibrium Networks. E.g. in WGANs, the stability and theoretical guarantees rely on the critic network being a 1-Lipschitz function. This stands in contrast to regularizers like L2 weight decay, which penalize the magnitude of weights without directly controlling the network's output sensitivity to input perturbations. While weight decay is a staple in real-valued neural networks, and applying it to quaternion NN learning should be straightforward (computing a quaternion norm should probably not require very important considerations for the quaternion domain), Lipschitz-based approach offers direct control over the network's smoothness, and finds use in the aforementioned domains (generalization error control, WGANs, etc.)
>
> But are there alternatives to what we do in Section 4 with Lipschitz bounding, specifically targeting Quaternionic Convolution? Works discussing power methods in the context of quaternion matrices do exist, but as far as we know they either focus on the right spectrum (while we show that we need the left eigenvalues / the right spectrum) e.g. [Li et al, “On the power method for quaternion right eigenvalue problem”, Journal of Computational and Applied Mathematics (JCAM) 2019] or make assumptions about the matrix being Hermitian [Jia et al., “A new structure-preserving method for quaternion Hermitian eigenvalue problems”, JCAM 2013] and, more importantly, they do not take into account the intricate structure of the matrix that is induced by quaternionic convolution e.g. [Grasucci et al. “Quaternion Generative Adversarial Networks”, 2021]. The operator A of quaternionic convolution is of course linear, hence a study about Lipschitz constant bounding can focus on A, but we have discussed in the paper that this matrix can very quickly become too big to even fit in memory. While there are methods that focus on this for real-valued convolution, notably [Sedghi et al., “The singular values of convolutional layers”, ICLR 2018] (which is an inspiration for much of Section 4) or the much more recent [Grishina et al., “Tight and Efficient Upper Bound on Spectral Norm of Convolutional Layers”, ECCV 2024], the field regarding adaptations or novel methods in the quaternionic domain is rather poor. Hence, to the best of our knowledge, straightforward adaptations of power methods have not been proposed in past literature [Altamirano-Gomez et al., “Quaternion Convolutional Neural Networks: Current advances and Future Directions”, 2024]. Furthermore, a naive implementation would have to solve fundamental issues in the quaternionic domain (but non-existent in real or complex domains), like taking into account that there are left and right spectra for each matrix, and these are completely (?) unrelated between themselves. Of course, this does not mean that better methods are not possible. On the contrary, we believe that the field is filled with interesting directions of research, like the one concerning power methods, implied by your comment, or adapting methods built on different rationales (see e.g. Table 1 of Grishina et al., ECCV 2024].
>
> We have modified the original text to accomodate for making reference to other regularizers, and other uses of regularization in the beginning of Section 4. Please let us know if you think that this modification should be extended and/or more parts of this discussion should be part of the paper text. Thank you again for you comment!

---

> ### Comment · Reviewer_3wXS · 2025-10-08
> **Thanks for author's rebuttal**
>
> Thanks for the author's rebuttal. So far, I believe my concerns have been addressed. I appreciate the efforts made by the authors in this regard. But due to my limited background and experience, I want to hear more of other reviewer's feedback.

---

> > ### Author Response · Authors · 2025-10-09
> >
> > Thank you for your response. We are very glad that you agree that we have addressed your concerns. Please let us know if and whether you want us to focus on a specific point in our rebuttal, or if you feel that a particular part of our text should be better clarified. We are in any case at your disposal.

---

### Comment · Action_Editor_dze6 · 2025-09-11
**Discussion**

Dear Authors, dear Reviewers,


This paper has received three high-quality reviews, and therefore the discussion should start in earnest.

It appears that the reviewers are **generally positive about the originality and interest of the method**. However, there are significant concerns to be addressed by the authors.

I list the ones which appear most severe (in decreasing order of severity) in my opinion, but the authors are encouraged to write a full rebuttal to all comments by all reviewers.


1 (reviewers 4wny and pLGg) The **proofs** and notation appear a little poorly polished. Reviewer 4wny identified concrete problems including redundant or invalid notation, a proposition which is missing a proof, and improperly delineated proofs. Reviewer pLGg gave many examples which demonstrate improper polishing of the mathematical notation. Since the field is a little out of scope by ML standards, it is especially important that the proofs are well polished and understandable to an audience without a proficient understanding of quantum mechanics. Writing down a statement without explicitly stating whether the statement is "to be proved later", "obvious" or "known" are not acceptable.  If you plan on using a citation instead of a proof for proposition 3.8, you should cite the specific theorem number and page, and it should match your own result *exactly* (except for notational permutation) without requiring additional deduction.


2. (reviewer QEr5) Consider incorporating another version of Figure 3 with the standard deviation instead of multiple curves for multiple random seeds. Personally, I would include both the current version and such an improved version, run **on more than the current number (4) of random seeds (try 100 if possible, otherwise at least 10)**. This is because looking at the individual seeds do show that there is significant deviation from the mean which may be of a fat-tail type rather than something which can be captured in the variance.

3. QEr5, pLGg: please compare the method to other regularization strategies, and elaborate on whether you are actually calculating the full SVD instead of relying on power methods. If the clipping is such that you aim to make the final weights significantly low rank ($r\ll w$ if $w$ is the width), you should certainly be using a more effective method than calculating the full spectrum.


@reviewers: if you feel that some of the other reviewers' concerns are already addressed in the original submission, feel free to come to the authors' defence (or agree with the reviewers) and begin discussing amongst yourselves as well.



Looking forward to hearing everyone's further inputs!

Best regards,

AE

---

> ### Author Response · Authors · 2025-09-21
>
> Dear,reviewers, dear Action Editor,
>
> Thank you so much for the time you took to read our paper thoroughly, and provide us with your review. We have done our best to address your comments, and improve the paper as much as we could in the time window of the rebuttal.
>
> More specifically, concerning the summary provided by the Action Editor:
>
>  * 1. (reviewers 4wny / pLGg) We have now supplied proofs for all propositions and corollaries, modified / improved notation as per the suggestions of the reviewers.
> * 2. (reviewers QEr5) We have improved all Figures and Tables with standard deviation bars on curves and figures, and the new results were all done following the same standard.
> * 3. (reviewers QEr5, pLGg) Concerning regularization, we responded by elaborating on the points raised by the reviewers. As one of the reviewers suggested (pLGg), we wrote about the more extended scope of application of Lipschitz constant bounding, which really improves considerably the importance of the result. Concerning other regularization strategies, we have elaborated about relations and comparisons to other schemes of regularization like weight decay (reviewer QEr5) and compared against the landscape of approaches such as power methods in the context of the quaternion domain (reviewer pLGg). We have also clarified that, as much as we believe that the application of Section 4 represents a real, practical scenario that our results enable, we again stress that it is only a “proof of concept", in the sense that the most important contribution of our paper lies in the content of Section 3, with the propositions that discuss Quaternion Convolution and its link to the Quaternion Fourier Transform. In any case, the method of Section 4 can compute a nuanced image of the eigenstructure of the convolutional operator, and supply singular values in a ‘single iteration’, in the sense that practically all that is required is computation of the QFT to obtain all left eigenvalues in one go, and perform N inexpensive QSVDs (of 2x2 matrices).
>
> Except from these points, we have tried to address all other concerns raised. We apologize in advance in the event that you think that an answer has been inadequate; let us know and we will try to fix the problem.
>
> Aside from answering the reviewers’ concerns with comments within the openreview rebuttal submission system, we have complied with revising our manuscript to reflect suggested changes. Changes in-text were made to appear in blue colour, in order to facilitate comparison to the original submission.
>
> Feel free to let us know if you think that the result and responses fit your requirements, or whether you would suggest further minor or major modifications and/or clarifications on our behalf.

---

### Decision · Action_Editor_dze6 · 2025-10-21

**Recommendation:** Accept with minor revision

**Additional Comments:**

The reviewers are **unanimously positive** in their final recommendations, and the paper's **contributions** appear **sound** and could even be seen as *pioneering a new niche class of models*. However, I would recommend the authors make one last attempt at making the paper reader friendly in the final camera-ready revision.

Indeed, much of the paper reads like a discourse on quaternions, and the way they are incorporated into the neural network learning procedure is not explained clearly. This concern of mine is shared by reviewer QEr5, whose question regarding the "optimization procedure" echoes my doubts. The authors have answered that  they "quaternionize” a standard neural network architecture  by "replacing the typical 3×3 convolution on the residual blocks with a depthwise quaternionic one" and that "the input convolution layer is not replaced by a quaternionic one." I don't feel this answer is enough, and in any case, it should be both included in, and elaborated upon, in a dedicated "architecture" subsection in the final manuscript. To the best of my understanding, the real-valued inputs are eventually fed through a quaternionic layer, which should be explained in mathematical detail. It is also worth discussing whether the fact that the input is real has any effect, and (importantly), why the authors have only chosen a subset of the layers to quaternionize, without going through a more detailed ablation study on the different possible architectures.




Minor typos:

Page 2: convolution is an ubiquitous… > convolution is a ubiquitous…

page 6: "exploring whether and to what extend do the well-known properties..."> "exploring whether and to what extent well-known properties..."

**Audience:**

Yes

**Audience Explanation:**

Although quaternionic networks are a niche topic, the machine learning community is expanding its scope in terms of topics. In addition, it is credible that some of the contributions of this work are laying the foundation for the use and study of quaternionic neural networks, providing a new model class some may find of interest.

**Claims And Evidence:**

Yes

**Claims Explanation:**

This paper establishes matrix forms for Quaternionic Convolution and the Quaternion Fourier Transform (QFT), and (importantly), demonstrates how to calculate the Lipschitz constant of quaternionic linear layers. In experiments, the authors show that their method for the calculation of the Lipschitz constant is faster than brute force calculation, and that imposing Lipschitz constraints on their quaternionic networks increases test accuracy.

Many of the reviewers initially complained of mathematical typos and lack of clarity. Although the paper is still a little hard to read for a machine learning audience, the reviewers unanimously agree that the revised version of this paper has done a decent job at clarifying assumptions and mathematical notations.